# Are LLMs Prescient?
# A Continuous Evaluation using Daily News as the Oracle

**Hui Dai** [1]   **Ryan Teehan** [1]   **Mengye Ren** [1]

## Abstract

Many existing evaluation benchmarks for Large Language Models (LLMs) quickly become outdated due to the emergence of new models and training data. These benchmarks also fall short in assessing how LLM performance changes over time, as they consist of a static set of questions without a temporal dimension. To address these limitations, we propose using future event prediction as a continuous evaluation method to assess LLMs' temporal generalization and forecasting abilities. Our benchmark, Daily Oracle, automatically generates question-answer (QA) pairs from daily news, challenging LLMs to predict "future" event outcomes. Our findings reveal that as pre-training data becomes outdated, LLM performance degrades over time. While Retrieval Augmented Generation (RAG) has the potential to enhance prediction accuracy, the performance degradation pattern persists, highlighting the need for continuous model updates. Code and data are available at https://agenticlearning.ai/daily-oracle.

## 1. Introduction

Traditional Large Language Model (LLM) benchmarks are often static, and do not reflect real-world information that evolves over time. This presents two significant challenges. First, as LLMs are updated, there is a risk that static benchmarks become outdated and more vulnerable to data leakage, where their content might end up in the training data of newer models. This undermines the reliability of performance assessments on these benchmarks (Sainz et al., 2023; Xu et al., 2024; McIntosh et al., 2025; Li & Flanigan, 2024). Second, static benchmarks often lack temporal in-

[1]New York University. Correspondence to: Hui Dai <hd2584@nyu.edu>, Ryan Teehan <rst306@nyu.edu>, Mengye Ren <mengye@nyu.edu>.

*Proceedings of the 42nd International Conference on Machine Learning*, Vancouver, Canada. PMLR 267, 2025. Copyright 2025 by the author(s).

formation, making it difficult to track models' performance variations over time (McIntosh et al., 2025). This creates a need for evaluation methods that always remain relevant and incorporate temporal dynamics.

Daily news provides a natural setting for continuous evaluation of LLMs. Since the world is constantly changing, a benchmark designed around forecasting the next day's news will never be out of date by construction. In addition to enabling continuous evaluation, forecasting is itself a longstanding challenge with significant implications across various domains, including healthcare, finance, and policy-making (Tetlock & Gardner, 2016; Dempsey et al., 2017; Gillingham et al., 2018; Lopez-Lira & Tang, 2023). While human experts have traditionally made such forecasts, machine learning models, particularly LLMs, have emerged as promising alternatives due to their ability to learn from vast and diverse corpora (Halawi et al., 2024; Ye et al., 2024; Yan et al., 2024). Several recent forecasting question-answer (QA) datasets have been developed (Jin et al., 2021; Zou et al., 2022; Zhang et al., 2024), however, they are limited in either size, scope, or they do not continuously keep pace with the rapidly changing world. More critically, the extent to which LLMs' predictive abilities change over time remains understudied.

In this work, we propose Daily Oracle—a continuous evaluation benchmark that uses automatically generated QA pairs from daily news to assess how the future prediction capabilities of LLMs evolve over time. The QA pairs are generated on a daily basis, consisting of True/False (TF) and Multiple Choice (MC) questions across various categories such as business, politics, and arts. Unlike traditional reading comprehension tasks, these QA pairs are designed to challenge LLMs to predict future events based on their own existing knowledge, effectively evaluating their temporal generalization and forecasting abilities.

We continuously evaluate various LLMs, both with and without access to a limited archive of news articles. Our experiments reveal that LLMs experience **significant performance degradation** between January 2020 and December 2024, with degradation becoming more pronounced before and after the models' knowledge cutoff dates. On average, performance drops by 21.55% on TF questions

and 11.33% on MC questions. While model performance can be improved with more recent news articles using Retrieval Augmented Generation (RAG) (Lewis et al., 2020), the **downward** trend persists, suggesting the challenge in maintaining its prediction ability over time.

To summarize, our key contributions are two-fold:

- **Continuous Forecasting Evaluation Benchmark**: We present Daily Oracle, the largest and most up-to-date forecasting dataset, composed of automatically generated QA pairs. This benchmark continuously evaluates LLMs' temporal generalization and future prediction abilities using daily news, ensuring relevance over time and offering a challenging evaluation framework.

- **Empirical Findings on Performance Degradation**: Since our benchmark provides new questions each day, we can study how model performance changes along the temporal axis. Our work effectively reveals a clear performance degradation pattern in LLMs' forecasting accuracy over time. Additionally, we study how this pattern changes as the LLMs are given access to updated knowledge up to different times. Surprisingly, we find that, **even when the model has access to recent information in an "open-book" setting,** it still experiences performance degradation. Moreover, the sheer degree of decline, along with its smoothness over time, was unexpected. On the one hand, this highlights the problems with outdated LLM pre-training data and on the other hand underscores the need for continuous model updating.

## 2. Related Work

**Temporal Generalization of LLMs.** Lazaridou et al. (2021) define temporal generalization as the ability of Language Models to generalize well to future data from beyond their training period. They demonstrate that Transformer-XL's performance deteriorates over time, evidenced by increasing perplexity when evaluated on post-training data. However, perplexity-based metrics have two main limitations: they cannot be applied to closed-source models lacking accessible logits, and increased perplexity does not necessarily indicate degraded performance on downstream tasks (Röttger & Pierrehumbert, 2021; Agarwal & Nenkova, 2022). Zhu et al. (2025) investigate temporal generalization using the Bits Per Character (BPC) metric. Similar to perplexity, BPC fails to capture higher-level performance on downstream tasks. In contrast, our work focuses on the downstream forecasting task, evaluating how well models understand world knowledge and make predictions. This approach offers a more reliable evaluation of temporal generalization with direct relevance to real-world applications and public interest.

**Dynamic QA Datasets.** While static QA datasets evaluate models on fixed knowledge snapshots, dynamic QA datasets incorporate a temporal dimension, allowing assessment of how models adapt to evolving information. Several dynamic QA datasets are proposed. Chen et al. (2021) construct TimeQA by using time-sensitive facts in WikiData with aligned Wikipedia passages to synthesize QA pairs. Zhang & Choi (2021) introduce SituatedQA by manually annotating temporally and geographically dependent questions. StreamingQA (Liska et al., 2022) and RealtimeQA (Kasai et al., 2024) are both dynamic benchmarks with QA pairs answerable from news articles. StreamingQA, however, does not provide continuous evaluation with always-relevant data. RealTimeQA does not address forecasting and is more like a plugin for a search engine, in the sense that it tests whether a model has updated its knowledge as facts change, rather than testing whether it can predict what will change given its knowledge of the past. FreshQA (Vu et al., 2024) contains a fixed set of human-written open-ended questions whose answers by nature can change based on new developments in the world, but is smaller and does not address forecasting. It is also updated weekly rather than daily. While all these datasets have some form of time-sensitivity like the Daily Oracle, they either do not provide continuous evaluation or do not evaluate forecasting capabilities, or neither.

**Forecasting Datasets.** Forecasting questions aim to assess a model's ability to predict the outcomes of future events based on its existing knowledge. Several datasets in the event forecasting field have been introduced. ForecastQA (Jin et al., 2021) used crowdworkers to collect 10,392 QA pairs from news articles. Zou et al. (2022) argue that the QA pairs from ForecastQA are often nonsensical or ambiguous since they are written by humans without forecasting expertise. They further introduce AutoCast, a forecasting dataset from popular human forecasting tournaments containing 6,707 QA pairs. While ForecastQA and AutoCast remain static, ForecastBench (Karger et al., 2025) regularly updates a set of 1,000 forecasting questions either sourced from forecasting markets or generated via fixed templates based on real-world event datasets. However, it still depends on users actively submitting new forecasting questions or maintaining the underlying datasets. In contrast, our Daily Oracle dataset is generated automatically from daily news articles, which means that it is never out of date, can easily grow its size automatically without additional inputs from human forecasters, and provides more comprehensive event coverage than human forecasting tournaments.

Similar to our generation method, TLB-forecast (Zhang et al., 2024) has an automatic forecasting QA generation framework using news articles. However, their dataset is constrained both temporally and topically, only containing cooperation and conflict events in Middle-Eastern countries

| Dataset | Continuous? | Interval | Forecast? | Size | Latest Update |
|---|---|---|---|---|---|
| TimeQA (Chen et al., 2021) | ✗ | None | ✗ | 20,000 | 2021 |
| SituatedQA (Zhang & Choi, 2021) | ✗ | None | ✗ | 4,757 | 2021 |
| StreamingQA (Liska et al., 2022) | ✗ | None | ✗ | 36,800 | 2021 |
| RealTimeQA (Kasai et al., 2024) | ✗ | None | ✗ | 1,470 | 2023 |
| FreshQA (Vu et al., 2024) | ✓ | Weekly | ✗ | 600 | 2024 |
| ForecastQA (Jin et al., 2021) | ✗ | None | ✓ | 10,382 | 2019 |
| AutoCast (Zou et al., 2022) | ✗ | None | ✓ | 6,707 | 2022 |
| ForecastBench (Karger et al., 2025) | ✓ | Biweekly | ✓ | 1,000 | 2024 |
| TLB-forecast (Zhang et al., 2024) | ✗ | None | ✓ | 6,604 | 2022 |
| FreshBench (Zhu et al., 2025) | ✓ | Unknown | ✓ | 2,769 | 2024 |
| Daily Oracle (Ours) | ✓ | Daily | ✓ | 31,510 | 2024[*] |

[*] Our experiments use the subset generated until December 2024. Daily Oracle itself remains active, continuing to generate new questions daily from 2025 onward.

*Table 1.* We compare Daily Oracle with existing benchmarks in the literature. For continuously updated datasets (e.g. Daily Oracle, FreshQA, FreshBench, and ForecastBench), "Interval" refers to the dataset update interval, and "Size" and "Latest Update" refer to the fixed data currently available. Our Daily Oracle benchmark is the only forecasting benchmark which is **continuously updated every day** using questions generated from daily news.

from 2015 to 2022. This restricts the dataset from evaluating more general event-prediction abilities. Furthermore, considering most of the powerful LLMs have been developed after 2020, the portions of the dataset covering earlier years may contain answers already seen during training. This prior exposure compromises the dataset's effectiveness as a forecasting benchmark. In contrast, our dataset spans a broader timeframe and covers more topics, offering a more comprehensive forecasting benchmark.

Note that none of the aforementioned datasets provide insights into how prediction ability changes over time. Zhu et al. (2025) introduce FreshBench, a forecasting dataset scraped from the Good Judgment Open platform, and also study temporal generalization. However, they report accuracy in a relatively short time window (from January 2023 to August 2024) with only 2,769 questions. While we observe a gradual performance decline in our dataset, they report significant fluctuations in model accuracy shortly after release. A closer look reveals key limitations of forecasting market–based questions for studying temporal generalization: they suffer from limited early coverage, inconsistent distribution over time, and reduced dataset size after filtering due to a high proportion of low-quality questions, making it difficult to reliably analyze temporal performance trends. In contrast, our automatically generated dataset has broad event coverage, consistent growth, and more uniform question quality over time.[1]

In order to clearly showcase the differences between our dataset and prior work, we highlight a few key features in Table 1. The Daily Oracle is the only benchmark which is **continuously updated** on a daily basis and evaluates

forecasting ability. Additionally, at the fixed size we use for analysis we provide significantly more evaluation examples than the other automatically updated benchmarks.

## 3. The Daily Oracle Dataset

In this section, we present Daily Oracle, a continuously updated QA benchmark of forecasting questions that are automatically generated from daily news. For our current analysis of LLM performance, we utilize a subset of the data consisting of 16,783 TF questions and 14,727 MC questions, covering a diverse range of forecasting topics, which are generated using daily news articles from January 2020 up until December 2024. However, our QA generation framework is continuous and updates daily. In Section 3.1, we describe our LLM-based dataset construction pipeline, detailing the data sources and the four-step construction process. Section 3.2 provides an analysis and general overview of the dataset. Lastly, in Section 3.3, we conduct a human evaluation, similar to our QA filtering process, to verify the quality of the generated QA pairs.

### 3.1. Dataset Construction

**Data Source.** Following Zou et al. (2022), we collect a large corpus of news articles from the daily-updated Common Crawl News Dataset (Nagel, 2016) with the `news-please` package (Hamborg et al., 2017). To further enrich our news dataset, we supplement it with daily scraped news using the `Newspaper3k` package.[2] We filter for mainstream sources—CBS News, CNBC, CNN, Forbes, and NPR. While our data collection and evaluation are performed daily, for this study we utilize a static news corpus

---

[1]See Appendix C for details on comparing LLM-generated and forecasting market datasets.

[2]https://newspaper.readthedocs.io/en/latest/

with 1,246,973 English articles spanning January 2019 to December 2024. This corpus is also used for the constrained open-book evaluation setting in Section 4.1.

**LLM-based Construction Process.** QA pairs are generated from articles published between January 2020 and December 2024.[3] For each day, we select six articles for QA generation: three are chosen randomly, and three are selected from hot topics.[4] For each selected article, we then use LLM to generate two TF QA pairs and two MC QA pairs with the few-shot prompting technique.[5]

We adopt the methodology of Zhang et al. (2024), as their prompt design largely suits our setting. To further filter the QA pairs, we establish seven key criteria to ensure they qualify as valid forecasting questions, incorporating these into a *QA Filtering step*. The QA construction follows four steps, as illustrated in Figure 7:

(1) *Article Summary.* We generate a summary for each article, focusing on new events from the publishing date, instead of opinion articles discussing events from the past. This approach allows us to use the publication date as the resolution date of the generated question. Questions can then be regarded as valid forecasting questions since they are prior to the resolution date.

(2) *QA Generation.* After filtering out the articles that do not introduce new events, two TF questions and two MC questions are generated together with the answers per article. To ensure balance in the TF questions, we instruct the LLM to generate the first question with a "Yes" answer and the second with a "No."

(3) *Misleading Choices Generation.* For MC, we provide the article, its publishing date, and the QA pair to the LLM, which then generates three misleading choices.

(4) *QA Filtering.* We prompt the LLM to check seven principles: correctness of answers, non-answerability before the publication date, absence of information leakage, objectivity, inclusion of a clear temporal element, public interest, and non-obviousness of the answer. Each principle is scored with 0, 1, or 2 points, and we selected the questions that received at least 13 points in total. These principles are detailed in Appendix A.3.

We use GPT-3.5 (OpenAI, 2024a) or GPT-4o-mini (OpenAI, 2024b) for steps (1) and (4), while GPT-4 (OpenAI, 2023) or GPT-4o (OpenAI, 2024b) is utilized for steps (2) and (3) to ensure high data quality.[6]

---

[3]For news articles in 2019, we use them as the corpus for the constrained open-book setting.

[4]See Appendix A.2 for hot-topic selection details.

[5]See Appendix D for all the prompts we use.

[6]We use GPT-3.5 and GPT-4 until September 2024. After

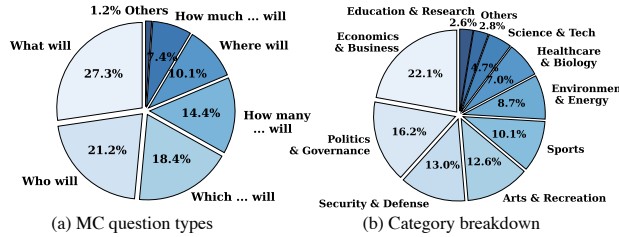

(a) MC question types      (b) Category breakdown

*Figure 1.* Pie charts showing (a) MC question type distribution and (b) question category distribution in Daily Oracle.

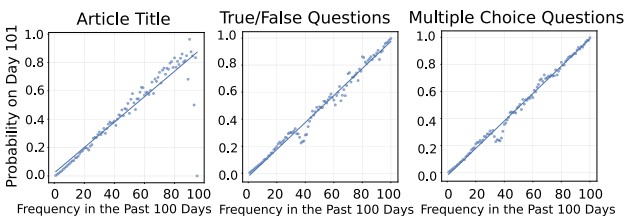

*Figure 2.* Following Anderson & Schooler (1991), we plot the probability of a word occurring in an **(left)** article title, **(middle)** True/False question, or **(right)** Multiple Choice question given how frequently it had appeared in one over the past 100 days, computed over our entire dataset. We fit a linear regression and show a linear relationship in each case ($R^2 = 0.843$, $0.982$, and $0.990$ for left, middle, and right respectively).

### 3.2. Dataset Analysis

**Summary Statistics.** At the time of writing this paper, the subset dataset we use from Daily Oracle consists of 16,783 TF and 14,727 MC QA pairs, covering the period from January 1st, 2020, to December 31st, 2024, with an average of 17.2 questions per day. Figure 1(a) shows that our dataset covers various MC question types, mainly starting with "What will" (27.3%), "Who will" (21.2%), and "Which ... will" (18.4%). Figure 1(b) provides a breakdown of the categories, highlighting our dataset's broad coverage. The categorization of each question is determined using GPT-3.5, based on the prompt from Halawi et al. (2024). Examples of QA pairs are shown in Table 2.

**Past and Future Information Usage.** Each question in Daily Oracle implicitly requires the model to retrieve relevant knowledge. How do these requirements change day by day over the course of our benchmark? Anderson & Schooler (1991) explored similar patterns in human information environments. Inspired by their work, Figure 2 examines whether a word's frequency over the past 100 days

---

October 2024, we switch to GPT-4o-mini and GPT-4o, as they are both more cost-effective and more powerful.

*Table 2.* Daily Oracle Example Questions and Answers.

| Type | Category | | Question and Answer |
| --- | --- | --- | --- |
| TF | | Politics & Governance | – Will the prosecution's key witness in the New York hush money trial in April 2024 be someone other than Michael Cohen? –**No.** |
| TF | | Politics & Governance | – Will the House Energy and Commerce Committee vote unanimously to advance a bill that could potentially ban TikTok if ByteDance does not sell the app by March 2024? –**Yes.** |
| MC | What | Science & Tech | – What will be the starting price range for the Google Pixel 8a as of May 2024? A.$599–$649 B. $199–$249 C. $750–$800, D. $499–$559. –**D.** |
| MC | Who | Sports | – Who will go on the injured list before the New York Mets' game on May 29, 2024? A. Pete Alonso B. Edwin Diaz C. Jeff McNeil D. Francisco Lindor –**B.** |
| MC | Which | Arts & Recreation | – By May 2024, on which streaming service will "The First Omen" become available for subscribers? A. Disney+, B. Hulu, C. Amazon Prime Video, D. Netflix –**B.** |
| MC | How many | Science & Tech | – How many U.S. states will the path of totality cross during the total solar eclipse on April 8, as reported by February 2024? A. 15 B. 10 C. 20 D. 6 –**A.** |
| MC | Where | Healthcare & Biology | – Where will the second known U.S. case of bird flu in a human be reported by March 2024? A. California, B. Texas, C. New York, D. Florida –**B.** |
| MC | How much | Economics & Business | – How much will Apple, Inc. (AAPL) be up year-to-date by the end of June 2024? A. Up 149.5% B. Just over 19% C. 9.7%. D. 27%. –**C.** |

predicts its occurrence the next day—e.g., if many questions concern the unemployment rate, will this trend continue?

We analyze this relationship for words in the titles of the articles we use to generate questions as well as in the text of the TF and MC questions themselves. Past frequency is computed by checking, for each day in the 100 day window, if a word has occurred in any article title (so, the maximum frequency is 100). We find that there is a linear relationship between the frequency of usage in the past 100 days and the probability of occurrence on the 101st day in all cases, replicating Anderson & Schooler's findings for *New York Times* headlines. This indicates that past information usage strongly predicts future retrieval needs, suggesting a temporal structure in the knowledge demands of our benchmark.

### 3.3. Human Evaluation

To assess the quality of our LLM-based filtering method, we randomly sample 30 TF and 30 MC QA pairs and ask 4 human annotators to evaluate them using the seven principles outlined in the *QA Filtering* step in Section 3.1. We evaluate the consistency among annotators using Fleiss' Kappa (Fleiss, 1971), which yields an average inter-rater agreement score of 0.26, indicating fair agreement. We then compute the human consensus score as the average of human scores and compare it to LLM-assigned scores, finding an average accuracy of 89.52% across the 7 principles. For final QA pair acceptance (i.e., threshold above 13 points), the LLM and human consensus scores demonstrated an accuracy of 85.00%, further supporting the reliability of our LLM-based filtering approach. A detailed breakdown of human evaluation metrics is provided in Appendix A.4.

## 4. Experiments

We first introduce three evaluation settings in Section 4.1: 1) no access to external information, 2) access to retrieved recent news articles, and 3) access to gold articles. Section

4.2 presents the results, and Section 4.3 provides deeper insights into the observed degradation patterns.

### 4.1. Experimental Setup

**Closed-Book Setting.** We evaluate various LLMs on Daily Oracle to assess their understanding of real-world events and temporal generalization abilities, i.e., how accurately LLMs can answer forecasting questions based on the knowledge they learned from their training data. Our evaluation differentiates between two scenarios based on the question's resolution date and model's knowledge cutoff date: (1) *Pre-Knowledge Cutoff Questions:* These questions have resolution dates before the model's knowledge cutoff, testing the model's understanding of past events. (2) *Post-Knowledge Cutoff Questions:* These have resolution dates after the knowledge cutoff, requiring models to predict future events and test their forecasting and temporal generalization abilities.

**Constrained Open-Book Setting.** In addition to a closed-book evaluation, we explore the constrained open-book setting: how access to news articles up to different time cutoffs influences LLM performance using RAG (Lewis et al., 2020). We introduce the concept of the RAG cutoff (*R-Cutoff*), which limits the latest accessible date for retrieving articles. To prevent the models from leveraging information beyond the resolution date, for any question with a resolution date ($d_{\text{res}}$), the accessible articles span from January 1st, 2019 (the start of our news corpus) up to whichever comes first between the day before the resolution date and the RAG cutoff date ($d_{\text{R-Cutoff}}$). Formally, the accessible date range is $[01/01/2019, \min(d_{\text{res}} - 1, d_{\text{R-Cutoff}})]$. Following prior work (Jin et al., 2021; Zou et al., 2022; Zhang et al., 2024), we employ BM25 (Robertson et al., 1995) as the retriever and select the top 5 articles relevant to each question. We truncate each retrieved article to a maximum length of 512 words. These articles are then incorporated into the input prompts to serve as additional information.

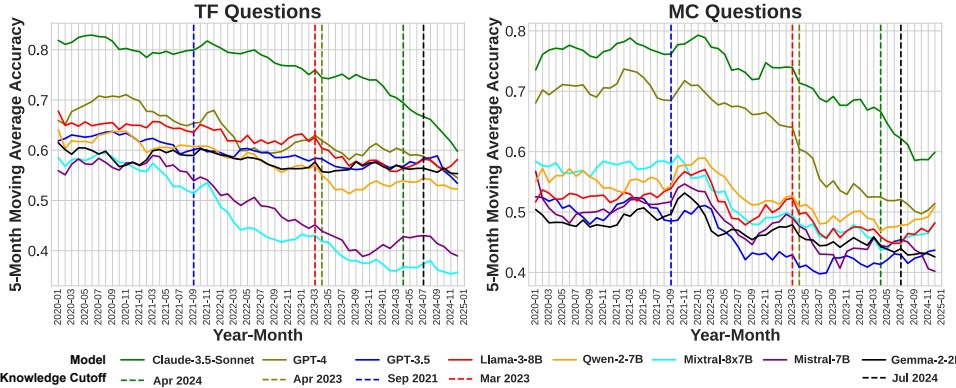

*Figure 3.* Results for the closed-book setting. We plot the 5-month moving average accuracy for TF and MC questions across various models, showing LLMs' performance degradation in future event prediction.

*Table 3.* For evaluating various LLMs with different knowledge cutoffs (K-Cutoffs), we show the yearly average accuracy (calculated as the average across months) from 2020 to 2024, along with the average YoY accuracy change (%) before the knowledge cutoff date (Pre-Cutoff), after the knowledge cutoff date (Post-Cutoff), and the overall average YoY accuracy change across all months (Avg).

| | LLM | K-Cutoff | Average Yearly Accuracy (%) | | | | | Average YoY Accuracy Change (%) | | |
|---|---|---|---|---|---|---|---|---|---|---|
| | | | 2020 | 2021 | 2022 | 2023 | 2024 | Pre-Cutoff | Post-Cutoff | Avg |
| **TF** | Claude-3.5-Sonnet | Apr 2024 | 81.21 | 79.88 | 78.05 | 74.38 | 64.29 | -4.77 | -12.41 | -5.58 |
| | GPT-4 | Apr 2023 | 69.68 | 66.41 | 60.36 | 60.54 | 56.90 | -5.83 | -1.96 | -4.75 |
| | GPT-3.5 | Sept 2021 | 62.86 | 60.12 | 59.36 | 57.11 | 56.09 | -4.33 | -3.43 | -2.84 |
| | Mixtral-8x7B | Unknown | 57.83 | 52.69 | 43.09 | 39.34 | 36.01 | – | – | -10.78 |
| | Mistral-7B | Unknown | 57.57 | 54.65 | 48.22 | 41.35 | 40.92 | – | – | -7.75 |
| | Llama-3-8B | Mar 2023 | 65.06 | 64.24 | 62.35 | 58.68 | 56.99 | -1.95 | -6.50 | -2.97 |
| | Qwen-2-7B | Unknown | 62.42 | 60.18 | 57.67 | 53.39 | 53.04 | – | – | -3.75 |
| | Gemma-2-2B | Jul 2024 | 58.71 | 59.31 | 57.64 | 56.61 | 55.79 | -1.41 | -3.68 | -1.04 |
| **MC** | Claude-3.5-Sonnet | Apr 2024 | 76.86 | 77.67 | 74.32 | 69.37 | 61.84 | -6.26 | -11.78 | -5.03 |
| | GPT-4 | Apr 2023 | 70.60 | 70.62 | 66.76 | 56.36 | 51.63 | -4.23 | -18.54 | -7.04 |
| | GPT-3.5 | Sept 2021 | 50.27 | 50.40 | 44.38 | 41.45 | 43.09 | 0.14 | -0.31 | -3.08 |
| | Mixtral-8x7B | Unknown | 57.38 | 56.97 | 50.76 | 47.10 | 46.31 | – | – | -4.68 |
| | Mistral-7B | Unknown | 50.07 | 52.36 | 48.06 | 44.40 | 42.99 | – | – | -2.82 |
| | Llama-3-8B | Mar 2023 | 52.44 | 54.18 | 50.66 | 47.94 | 46.95 | -2.21 | -1.25 | -2.30 |
| | Qwen-2-7B | Unknown | 55.28 | 55.93 | 53.44 | 49.77 | 49.37 | – | – | -2.35 |
| | Gemma-2-2B | Jul 2024 | 47.87 | 50.71 | 46.81 | 45.20 | 43.28 | -4.46 | -4.07 | -1.98 |

**Gold Article Setting.** We further include a setting where models are provided direct access to the gold article, from which the question is generated.[7] This transforms the forecasting questions into reading comprehension ones, which can also access LLMs' general question-answering capabilities. Achieving high accuracy here ensures that the questions from our Daily Oracle dataset are answerable.

**Metrics.** Accuracy score is used as the evaluation metric. Though LLMs are tested daily, to show clearer trends, we plot the monthly performance in Figure 3, and apply a 5-month moving average to smooth the curve. We also report yearly averages and average year-over-year (YoY) accuracy change before and after models' knowledge cutoff dates in Table 3. Additionally, despite prompting the models to avoid responses like "I cannot predict the future" and

instead provide definitive answers, there are cases where such refusals still occur. The refusal rates are provided in the Appendix B.2, and these cases are counted as incorrect to ensure comparability across model results.

### 4.2. Main Results

**Results for the Closed-Book Setting.** Figure 3 and Table 3 present our primary results for the closed-book setting. The "Avg" column in Table 3 shows the average YoY accuracy change of all months, revealing a clear degradation in performance over time across all models on both TF and MC questions. When comparing accuracies from the beginning to the end of the evaluation period, we observe that, on average, the models' performance declines by 21.55% on TF questions (from 64.68% to 50.74%) and by 11.33% on MC questions (from 58.30% to 51.69%). This indicates that while LLMs demonstrate certain abilities to understand real-world events and make predictions, they struggle to

---

[7]See Appendix B.5 for a case example of evaluating LLMs under all three different settings.

maintain these abilities.

Notably, the average YoY accuracy declines provide further insight. Before the knowledge cutoff, the average YoY decline across all models was relatively moderate. However, post-knowledge cutoff, we observe steeper declines in many models, with GPT-4 showing the most drastic drop in MC performance, declining by 18.54%, compared to just 4.23% before the cutoff. This contrast highlights that while LLMs manage to retain a baseline of past knowledge with small degradation, their ability to forecast future events deteriorates much more rapidly as they move beyond their training data, struggling with temporal generalization.

Among different models, Claude-3.5-Sonnet (Anthropic, 2024) significantly outperforms all others, while GPT-4 excels in MC questions but its performance in TF is not as remarkable as in MC. GPT-3.5, Qwen-2-7B (Yang et al., 2024) and Llama-3-8B (Dubey et al., 2024) show smaller temporal declines than GPT-4 in both TF and MC questions. Interestingly, Mistral-7B (Jiang et al., 2023) and Mixtral-8x7B (Jiang et al., 2024) show the most pronounced drops in TF accuracy, with scores falling below the random baseline 50% due to increased answer refusals, as shown in Figure 9. Gemma-2-2B (Team et al., 2024) exhibits the most consistent performance with the smallest average YoY decline, likely due to its more recent knowledge cutoff date.

**Results for the Constrained Open-Book Setting.** In Figure 4, we present the results of the constrained open-book setting, with Mixtral-8x7B on TF questions and Llama-3-8B on MC questions across different RAG cutoff dates.[8] For Mixtral-8x7B, as the RAG cutoff dates extend to closer to the resolution dates, we observe a clear improvement in performance, indicating the model benefits from increasingly updated information retrieval. However, there are noticeable performance drops immediately after each RAG cutoff date when compared to providing information up to the day before the resolution date. This highlights the importance of keeping up-to-date information for optimal RAG performance. Interestingly, RAG does not uniformly enhance performance. Llama-3-8B may perform worse than the closed-book setting when the RAG cutoff is prior to the knowledge cutoff dates, suggesting outdated information may negatively impact performance. Conversely, for more recent RAG cutoff dates that extend beyond the knowledge cutoff, significant performance improvements are observed (as illustrated by the curves with cutoffs in September 2023 and March 2024). Notably, across all different RAG cutoffs, the overall performance decline pattern persists, likely due to outdated internal representations and the model's inherent knowledge limitations.

**Results for the Gold Article Setting.** Figure 5 shows that when given access to the gold articles from which the questions are generated, LLM performance can approach around 90%, demonstrating the answerability of Daily Oracle.[9] However, most of the models still show declining trends. This is noteworthy because, ideally, LLMs are expected to achieve consistent accuracy regardless of the article's publication date when answers are directly accessible. However, the outdated representations hinder their ability to consistently generate correct answers, even in a reading comprehension setting.

### 4.3. Discussion

**LLMs' Performance Degradation Pattern Over Time.** We observe LLMs' performance evolution patterns in Figure 3: (1) *Gradual Decline in the Recent Past:* In the months before the knowledge cutoff date, which we call the *recent past*, we observe a gradual decline in model performance, as seen in Llama-3-8B, GPT-4, and Claude-3.5-Sonnet, likely due to a lack of representation of recent news in the training data. (2) *Rapid Decline in the Near Future:* In the *near future*, which we define as the months following a model's knowledge cutoff date, sharp performance drops are observed in several models in MC questions. For instance, the decline in Claude-3.5-Sonnet and GPT-4 accelerates soon after their knowledge cutoffs. Most of the models, however, do not lose all the predictive power at once, as evidenced by the further decline into the farther future.

We explore this further by analyzing the slope of accuracy as a function of time. In Figure 6, we show how the slope changes as we fit a regression to an increasingly larger window of data, until we reach the full set of accuracies. Specifically, using the 5-month moving average of each model's accuracy on MC questions (visualized in Figure 3), we start by fitting a linear regression line on the first 10 months of data. We then add an additional month and compute a new regression on the larger window, repeating until we reach the final month, and applying an exponential decay weighting to past data to reduce the influence of distant observations. With this, we can analyze how the slope of our regression line changes as each month is added to the data. The slope in each case is negative after the cutoff data and for Claude-3.5-Sonnet, GPT-4, and Llama-3-8B, the slope eventually or immediately becomes more negative than it was at any point preceding the cutoff. Both Claude-3.5-Sonnet and Llama-3-8B have a crossover from positive to negative slope in late summer 2022, July and August, respectively, while GPT-4's seems to occur slightly earlier, in March of 2022. For GPT-3.5, GPT-4, and Llama-3-8B, the slope becomes

---

[8]Refer to Appendix B.4 for results of other models in the constrained open-book setting.

[9]Results for GPT-3.5 are provided and discussed in Appendix B.3, as this older model performs relatively poorly and including it on the same scale would obscure the trends of other models.

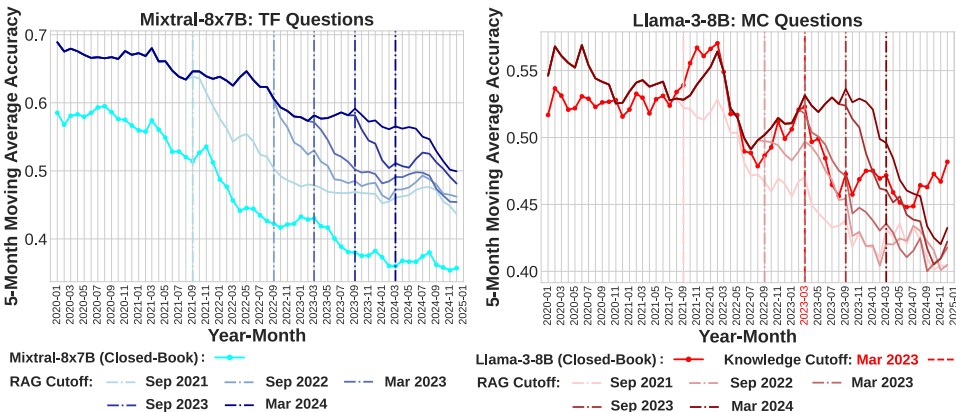

*Figure 4.* Results for the constrained open-book setting, evaluating Mixtral-8x7B on TF questions and Llama-3-8B on MC questions with different RAG cutoff dates.

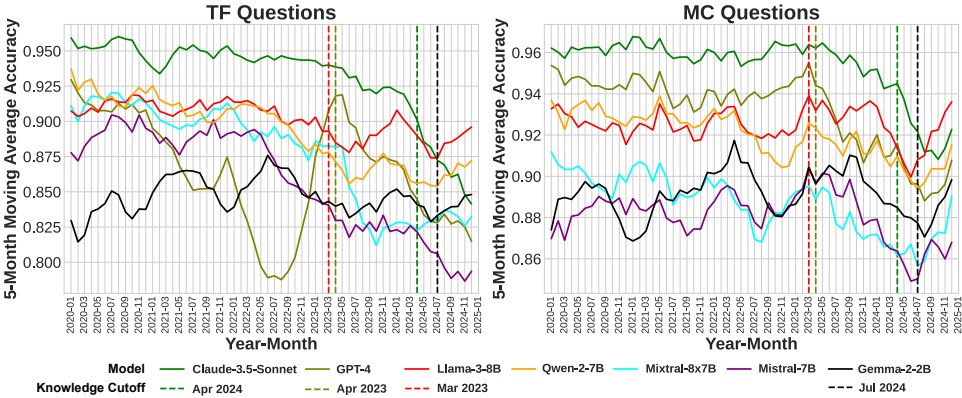

*Figure 5.* Results for the gold article setting. Most of the models struggle with temporal generalization, even when provided with gold articles containing the answers.

increasingly negative not long after the knowledge cutoff, giving evidence for a rapid decline in the *near future*. For example, Llama-3-8B's slope is around -0.06% per month near its cutoff in March 2023, but declines by more than 2 times to approximately -0.13% per month by the end of 2023. Likewise, the period preceding the cutoff shows a milder decline, with models like GPT-3.5 and Llama-3-8B exhibiting slightly negative but consistent slopes (approximately -0.01% and -0.06% per month respectively, over the four months leading up to the cutoff). This suggests a gradual decline in the *recent past*.

**Need for Continuous pre-training.** The overall decline trend may come from two sources, the missing knowledge of future and a lack of up-to-date language representation. The absence of relevant future information can lead to two outcomes: either the model makes uninformed or incorrect predictions, or, in some cases, more likely to refuse to answer altogether. We observe this latter behavior notably in Mistral-7B and Mixtral-8x7B, where refusal rates are

significantly higher compared to other models, as shown in Figure 9(b).[10] The lack of knowledge can be partially recovered with information retrieval, as seen in the constrained open-book and gold article settings. For instance, Figures 9, 10, and 11 show that Mixtral-8x7B's refusal rate drops from 14–28% in closed-book to 3–15% with open-book retrieval, and further to 0.5–4.2% with gold articles. However, accuracy still declines over time. Notably, the gold article setting provides an "upper bound" of open-book retrieval. The remaining performance drop despite full access to relevant information suggests that the models' internal representations are outdated. This indicates continuous pre-training of LLMs (Jang et al., 2022; Jin et al., 2022; Ke et al., 2022a;b; Yıldız et al., 2024) is still needed in the context of news event forecasting.

**TF & MC Comparison.** All models except for Claude-3.5-Sonnet struggle with TF questions, where the degrada-

---

[10]See Appendix B.2 for more discussion of the refusal behavior.

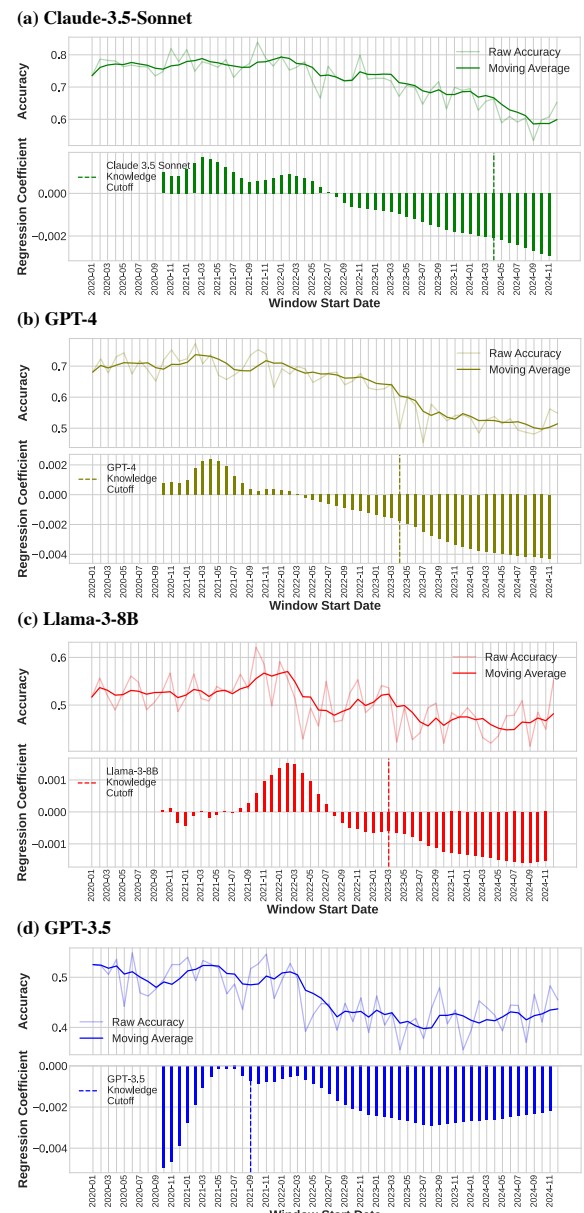

*Figure 6.* Coefficients for regressing accuracy on the MC questions against time, as the number of months grows. Using an initial window of 10 months, we progressively add data for additional months to our regression and plot the coefficient (slope) for the regression of accuracy against time. For our regression, we use the moving average of accuracy and apply exponentially decaying weights to older months (i.e., given a window of $k$ months, we weight $x_t$ with $\lambda^{k-t}$; in this case $\lambda = 0.995$).

tion trends towards the random baseline accuracy of 50%, indicating that predicting if a future event will happen or not can be sometimes challenging for LLMs. In contrast, on MC questions, models tend to perform much better than the random baseline at 25%. There are two potential reasons

that can explain the disparity. First, TF questions can be considered more open-ended than MC because the "No" answer contains other possible open-ended outcomes. Second, since the distractor choices are created by an LLM, they may not be as likely to happen as the true answer.

**Consistent Performance Decline After September 2021.**
Interestingly, Figure 3 reveals a higher rate of performance decline around September 2021, which is the knowledge cutoff date of GPT-3.5, across all models, particularly for MC questions. In contrast, performance remains relatively stable prior to this date. We hypothesize that this trend arises because the period up to September 2021 may be overrepresented in many pre-training corpora (Raffel et al., 2020; Gao et al., 2020; Kobayashi, 2018; Gokaslan & Cohen, 2019; Zhu et al., 2015; Rae et al., 2020; Tiedemann, 2016; Saxton et al., 2019), compared to more recent periods. Another potential cause of this imbalance is an increasing number of websites restricting access to web crawlers after the rise of ChatGPT (Longpre et al., 2024).

**Limitations.** On the data generation side, the generated questions as well as the distractor answers could contain biases from an outdated LLM, making the benchmark less reliable in the long run unless we upgrade the models. Additionally, generating questions from news articles can introduce bias by focusing only on events that have definitively occurred, overlooking potential events that never occur and thus never appear in the news. On the evaluation side, our paper proposes the continuous evaluation benchmark but at the time of the writing there isn't a long enough time horizon on each model, especially after the cutoff dates, for a thorough analysis. Ideally, we would like to analyze the relation between the effect of knowledge and RAG cutoff dates but the trend seems to be weak within the time horizon available.

## 5. Conclusion and Future Work

We introduce Daily Oracle, a continuously updated QA benchmark leveraging daily news to evaluate the temporal generalization and future prediction capabilities of LLMs. Our experiments reveal that while LLMs maintain a degree of predictive power over future events, their prediction accuracy exhibits a significant smooth decline over time. Although RAG mitigates the effect of outdated knowledge, a strong and noticeable decline remains. Our findings in the gold article setting further emphasize the importance of disentangling missing knowledge from the lack of up-to-date representations. In the future, alongside maintaining Daily Oracle, we plan to incorporate a broader range of models and explore how continuous pre-training and efficient adaptation can address the performance degradation challenges presented in our work.

## Impact Statement

Daily Oracle serves as an up-to-date, continuous evaluation benchmark for assessing AI models' forecasting accuracy and temporal generalization. These abilities have broad applications in areas such as finance, healthcare, and policy-making. The performance drop we observe highlights the risk of outdated knowledge and the importance of continuous model updates to keep AI systems reliable. In the future, by assessing continuous model update strategies in Daily Oracle, the broader ML community can gain valuable insights into how to maintain AI systems that are relevant and well-informed on recent and upcoming events.

## Acknowledgment

This work was supported in part by the Institute of Information & Communications Technology Planning & Evaluation (IITP) with a grant funded by the Ministry of Science and ICT (MSIT) of the Republic of Korea in connection with the Global AI Frontier Lab International Collaborative Research. (No. RS-2024-00469482 & RS-2024-00509279) We also thank the Microsoft Accelerating Foundation Models Research program for providing Azure cloud compute credits for the LLM APIs. The compute was also supported by the NYU High Performance Computing resources, services, and staff expertise.

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

# Appendix

## A. Dataset Details

### A.1. Illustration of the dataset construction process.

Figure 7 shows how Daily Oracle is automatically generated as discussed in Section 3.1.

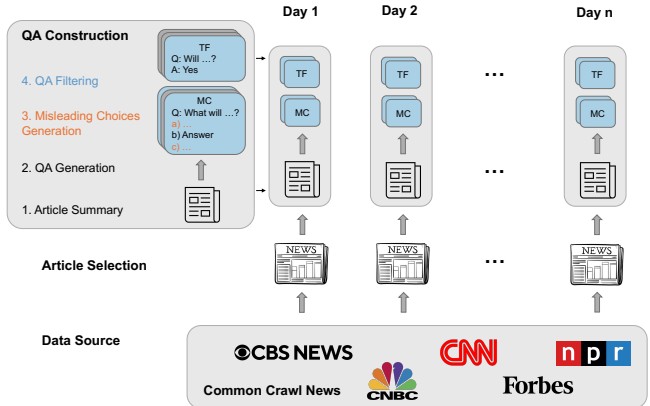

*Figure 7.* Data Construction Process of Daily Oracle.

### A.2. Details for Article Selection

We select daily articles that generate the QA pairs in two ways: (1) *Random Selection:* We randomly sample three articles each day. (2) *Hot Topic Selection:* To better capture daily events and reduce noise, we select three articles from the top three hot topics of the day. We identify these hot topics by applying the density-based clustering algorithm DBSCAN (Ester et al., 1996) to the new articles based on TF-IDF (Term Frequency-Inverse Document Frequency) representations, forming clusters of news articles for each day. We filter out chaotic clusters by removing those with low average in-cluster cosine similarity scores, which typically correspond to clusters containing a large number of diverse articles. The top three clusters, determined by size, are assumed to represent the most discussed events, i.e. hot topics, since larger clusters indicate more articles covering the same event. One article is picked randomly from each of the top three clusters.

### A.3. QA Filtering Principles

During the design stage of QA pair generation, we manually review the questions and identify seven key criteria to ensure the QA pairs qualify as valid forecasting questions. These principles guide the QA filtering step in the data construction process:

(1) *Correctness of Answers:* The answer must be factually accurate and fully aligned with the information in the given article.

(2) *Non-answerability Before the Publication Date:* Since we treat the article's publication date as the question's resolution date, the question should not be definitively answerable based on information available before the article's publication.

(3) *Absence of Information Leakage:* Questions must avoid revealing information that became known only after the article's publication, maintaining fairness for pre-publication evaluation.

(4) *Objectivity:* Both questions and answers must rely on objective facts, avoiding subjective ideas from the authors.

(5) *Inclusion of a Clear Temporal Element:* Questions must contain a specific and clear reference to time, avoiding vague phrases like "in the future" or "soon."

(6) *Public Interest:* The questions should address topics of broad public concern.

(7) *Non-obviousness of the Answer:* The answer should not be immediately predictable from the question and must provide new or non-trivial insights.

## A.4. Details for Human Evaluation

We assess the quality of our dataset by evaluating the effectiveness of our LLM-based evaluator in the *QA Filtering* step. Four human annotators independently review a randomly sampled subset of Daily Oracle, consisting of 30 TF and 30 MC QA pairs. They follow the same instructions used to prompt the LLM and evaluate each QA pair based on the seven filtering principles listed in Appendix A.3.

Table 4 presents the inter-rater agreement among human annotators and the agreement between human and LLM evaluators. The average Fleiss' Kappa of 0.26 indicates fair agreement among annotators. Among the seven principles, *Objectivity* exhibits the highest agreement (0.66), while *Non-Answerability Before the Publication Date* has the lowest (0.02).

Comparing human-assigned and LLM-assigned scores, the exact-match accuracy between the human consensus and LLM evaluations averages 89.52% across the seven principles, showing the effectiveness of our LLM-based filtering method. *Non-Answerability Before the Publication Date* shows the lowest agreement (83.33% accuracy), suggesting it is the most challenging principle for both humans and the LLM to evaluate consistently.

*Table 4.* The inter-rater agreement among four human annotators is evaluated using Fleiss' Kappa, while the agreement between human and LLM evaluators is measured through accuracy scores. We report metrics across seven *QA Filtering* principles using a sample of 60 randomly selected QA pairs.

| Metric | Human Agreement Fleiss' Kappa | Human vs. LLM Accuracy (%) |
|---|---|---|
| Correctness of Answers | 0.11 | 96.67 |
| Non-answerability Before the Publication Date | 0.02 | 83.33 |
| Absence of Information Leakage | 0.33 | 86.67 |
| Objectivity | 0.66 | 98.33 |
| Inclusion of a Clear Temporal Element | 0.21 | 90.00 |
| Public Interest | 0.18 | 88.33 |
| Non-obviousness of the Answer | 0.29 | 83.33 |
| Average | 0.26 | 89.52 |

## A.5. Distribution of Question Categories Over Time

In Figure 8, we provide the distributions of question categories for both TF and MC questions.

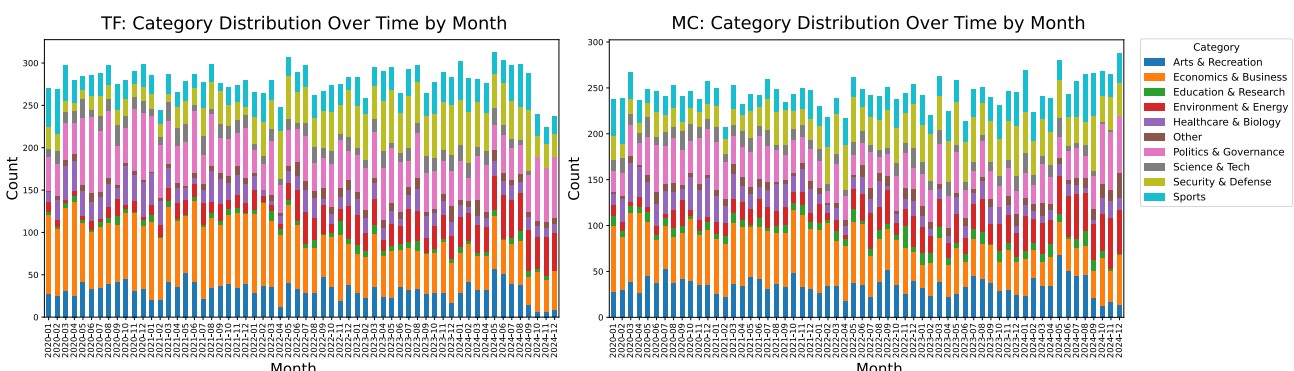

*Figure 8.* Question category distribution over time.

# B. Experiment Details

## B.1. Baseline Models Information

Table 5 lists the LLM model versions used in our experiments.

*Table 5.* Baseline model versions.

| Model | Model Version |
|-------|---------------|
| Claude-3.5-Sonnet | claude-3-5-sonnet-20240620 |
| GPT-4 | gpt-4-1106-preview |
| GPT-3.5 | gpt-3.5-turbo-0125 |
| Mixtral-8x7B | Mixtral-8x7B-Instruct-v0.1 |
| Mistral-7B | Mistral-7B-Instruct-v0.3 |
| Llama-3-8B | Meta-Llama-3-8B-Instruct |
| Qwen-2-7B | Qwen2-7B-Instruct |
| Gemma-2-2B | gemma-2-2b-it |

### B.2. Refusal Rates

Although the models are prompted to provide definitive answers rather than responding like "I cannot predict the future," some models still occasionally refuse to do so. Figures 9(b), 10(b), 11(b) show the refusal rates for the closed-book, constraint open-book, and gold article settings. In closed-book evaluation (Figure 9(b)), we can see that the refusal rates increase throughout the time for Mistral-7B in TF questions and Mixtral-8x7B in both TF and MC questions. Additionally, these two models exhibit notable refusal rates, with approximately 10–30% on TF questions and 1.5–8% on MC questions, resulting in their closed-book performances dropping below the random baseline of 50% in certain months, as shown in Figure 3. In comparison, Qwen-2-7B and Gemma-2-2B show relatively low refusal rates—<5% for TF and <2% for MC—while all other models have near-zero refusal rates for TF and <1% for MC.

The refusal behavior is likely influenced by alignment techniques, which discourage uncertain responses in the post-training stage. Although refusal rates contribute to lower accuracies for certain models, our results show that performance degradation trends persist even when refusals are excluded (Figures 9(a), 10(a), 11(a)). We consider refusal to answer an indicator of performance limitations in forecasting tasks, as it reflects the model's lack of actionable knowledge. When models are supplied with more up-to-date and relevant information, their refusal rates decrease (Figure 10(b)). This suggests that refusal is one example of the broader challenge of temporal generalization and reinforces the need for continual model updates or improved external knowledge integration.

### B.3. Results for GPT-3.5 in the Gold Article Setting

To more effectively illustrate the trends of other models at a suitable scale, we display GPT-3.5's performance in the gold article setting separately. As shown in Figure 12, this outdated model performs relatively poorly throughout. While its accuracy could improve with chain-of-thought prompting (Wei et al., 2022), we report its performance using the same prompt format as the other models for consistency in comparison. Nevertheless, the degrading trend can still be observed.

### B.4. More Results in the Constraint Open-Book Setting

Figures 13, 14, 15, 16, 17, 18, and 19 show the constrained open-book evaluation results for more models. Similar patterns are observed as discussed in Section 4.2. Specifically, for Claude-3.5-Sonnet, the constrained open-book performance lags behind its closed-book performance, likely because it already has robust representations of world events, suggesting that irrelevant or confounding retrieved information may degrade performance. This highlights the need for more careful RAG design in models that already possess robust world knowledge. GPT-3.5 is not included in the constrained open-book setting due to its unexpectedly poor performance in the gold article setting (Figure 12) and budget limitations. Additionally, due to budget constraints, open-book evaluations of proprietary LLMs (Claude-3.5-Sonnet, GPT-3.5, GPT-4) are conducted only up to September 2024, whereas other LLMs are evaluated through December 2024.

### B.5. An Example of Evaluating LLMs Under Different Settings

Figure 20 presents a case study demonstrating how Mixtral-8x7B responds to a question under different experimental settings. The model provides an incorrect answer in the closed-book setting. However, when supplemented with retrieved relevant articles or the gold article, it produces the correct answer.

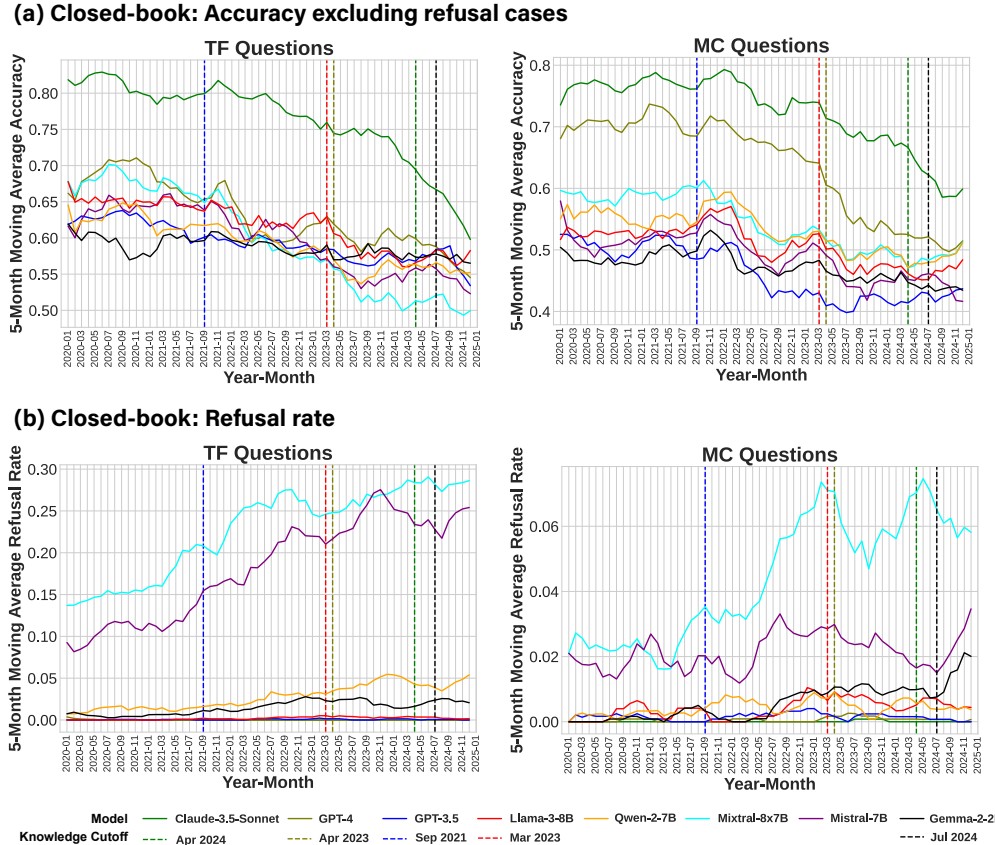

*Figure 9.* Accuracy excluding refusal rates and refusal rates under the closed-book setting. We plot the 5-month moving average refusal rates for TF and MC questions across different LLMs. We count refusal cases as incorrect both to maintain comparability across models and because failing to provide an answer when a prediction is expected represents an unsatisfactory outcome from the user's perspective.

## C. Comparing LLM-Generated and Forecasting Market Datasets

Online forecasting markets such as Metaculus and Polymarket allow users to submit questions and predict the outcomes of future events. A natural question arises: why do we focus on LLM-generated questions rather than sourcing from these markets, as the choice in prior work (Zou et al., 2022; Halawi et al., 2024; Karger et al., 2025)?

To answer this, we analyze the dataset from Halawi et al. (2024), which compiled 50,343 raw questions from five forecasting platforms, of which 21,149 were resolved. Of these, 82.64% are TF questions, 13.36% are MC questions, and the rest are free-response or numerical. After their quality filtering, only 5,516 TF questions remained. We observe that, due to a high proportion of low-quality questions in the raw data, sparse coverage in earlier years, and inconsistent distribution over time, performance trends derived from this dataset are substantially more volatile and harder to interpret than those based on our LLM-generated dataset.

**Lower Quality in Raw Questions.** Manual inspection confirms that a substantial portion of the raw dataset consists of low-quality questions, as also noted by the original authors. Examples include: *"Will I have a chess.com rating of >1300 ...?"* (personal), *"Will Jamaica beat Mexico?"* (missing a time frame), and *"Are there more disadvantages in AI than advantages?"* (ill-defined). From a random sample of 50 questions, only 28% were well-defined. Specifically, 26% lacked a clear time element, 20% were overly personal, and 26% were ill-defined. Importantly, only 5,516 out of 17,477 resolved TF questions were retained after their filtering—an acceptance rate of just 32%, which aligns with our own quality assessments.

**Limited Early-Year Coverage.** Figure 21 (left) shows that the coverage before October 2022 is sparse, averaging only 40 raw and 26 filtered questions per month. This scarcity limits the feasibility of longitudinal trend analysis, especially

**(a) Constraint open-book: Accuracy excluding refusal cases**

**(b) Constraint open-book: Refusal rate**

*Figure 10.* Accuracy excluding refusal rates and refusal rates for Mixtral-8x7B under the constrained open-book setting. (b) shows that the open-book refusal rate (blue curves) is lower than in the closed-book setting (cyan curve), indicating that access to more up-to-date and relevant information reduces model refusals.

across earlier model pre-training cutoffs. In contrast, our method supports high scalability and retrospective generation, allowing for uniform coverage across the full time range.

**Harder-to-Discern Trends.** To evaluate the impact of using forecasting market questions in our study, we run a closed-book evaluation on TF questions from both the raw dataset (16,089 questions) and the filtered subset (4,572 questions), starting from 2020-01 (the same start date as our dataset). Notably, the original data is imbalanced, with 61.03% "No" answers in the raw set and 64.28% in the filtered set. After balancing, we retain 12,438 questions from the raw data and 3,232 from the filtered set. As shown in Figure 21 (right), neither the raw nor filtered datasets reveal a clear performance trend—model accuracy fluctuates significantly over time. We believe this is due to several factors:

- **Lower data quality**: Approximately 70% of raw questions exhibit quality issues. While the overall dataset sizes are comparable (13,744 in ours vs. 12,438 in the raw market dataset), the difference in quality introduces additional noise, making trends harder to detect.

- **Limited early coverage**: Even within the filtered dataset, sparse early coverage and inconsistent monthly volume increase variance and reduce the reliability of time-based trends.

- **Confounding factors**: We argue that human-submitted questions introduce more confounding factors than automatically generated ones. Figure 22 shows the distribution of data sources and question categories varies significantly across time (e.g. more sports-related questions in later periods). Human-written questions also may differ widely in style and difficulty, making them harder to control for consistency. In contrast, as shown in Figure 8, our dataset maintains relatively stable distributions over time.

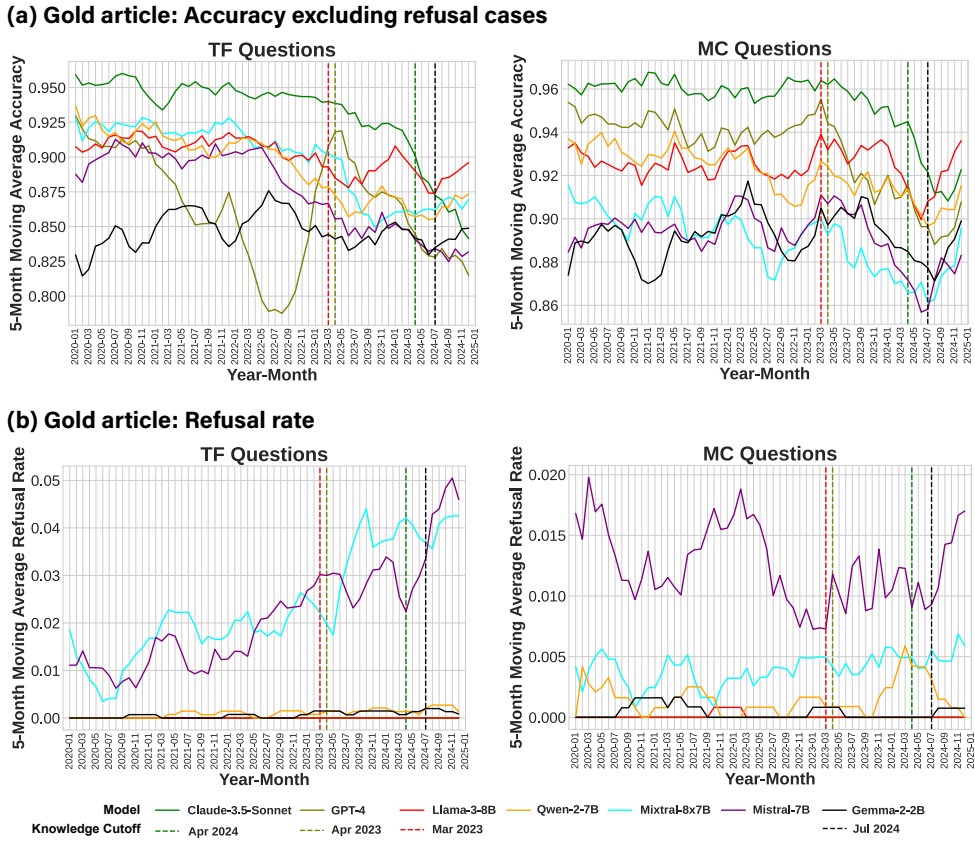

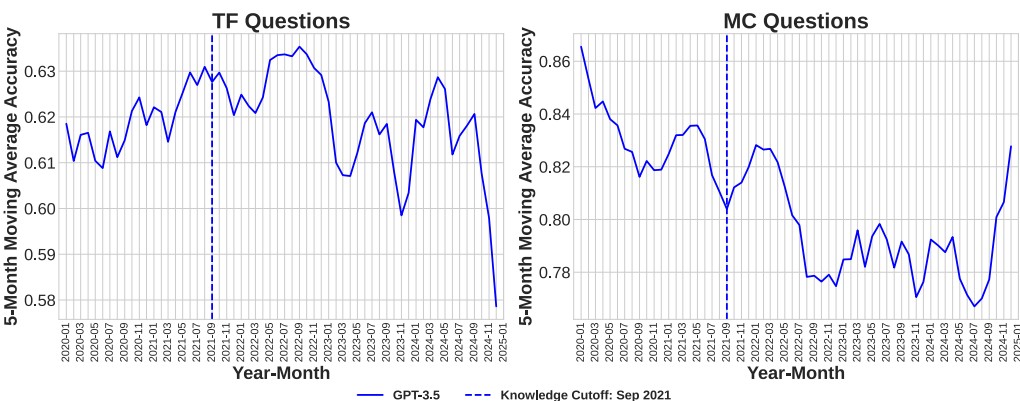

*Figure 11.* Accuracy excluding refusal rates and refusal rates under the gold article setting.

*Figure 12.* Results for GPT-3.5 in the gold article setting. Compared to other models achieving around 0.9 accuracy, GPT-3.5 performs worse in both MC questions and, more notably, in TF questions.

Thus, while we do not claim that LLM-generated questions are of inherently higher quality, we argue that our dataset is better suited for analyzing performance trends over time, due to its scalability, stylistic uniformity, stable category distribution, and reduced susceptibility to human-authored confounders. Moreover, if one sources questions from forecasting markets, the dataset update frequency is dependent on whether people are still actively submitting high-quality forecasting questions to the platform. In contrast, our approach enables daily updates and more comprehensive event coverage, making it a valuable complement to human-curated forecasting benchmarks.

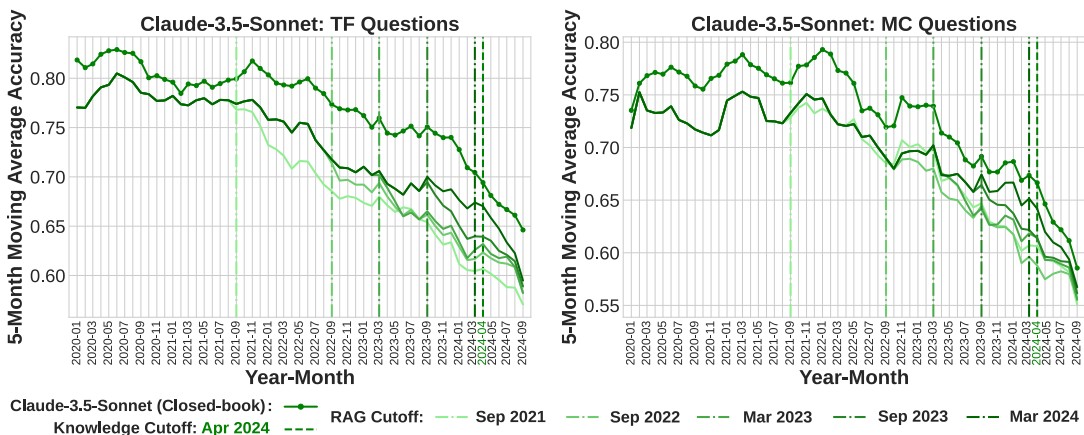

*Figure 13.* Results for Claude-3.5-Sonnet in the constrained open-book setting.

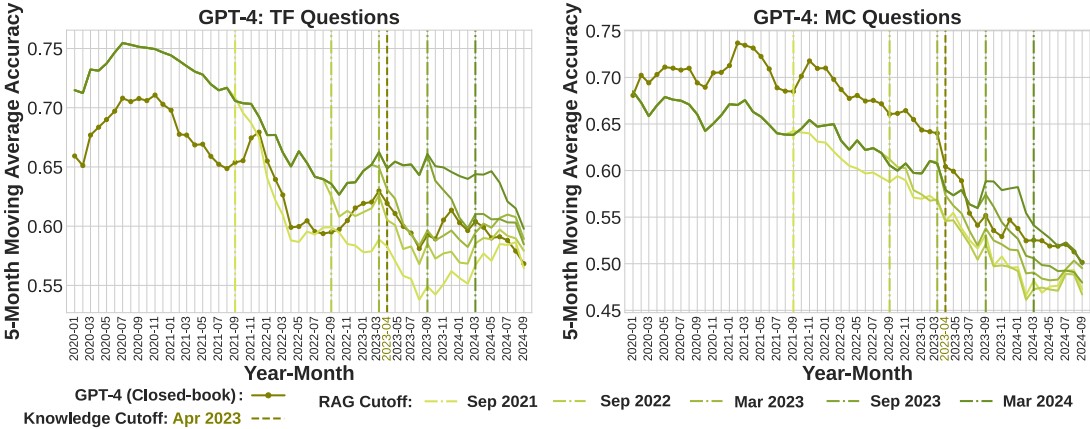

*Figure 14.* Results for GPT-4 in the constrained open-book setting.

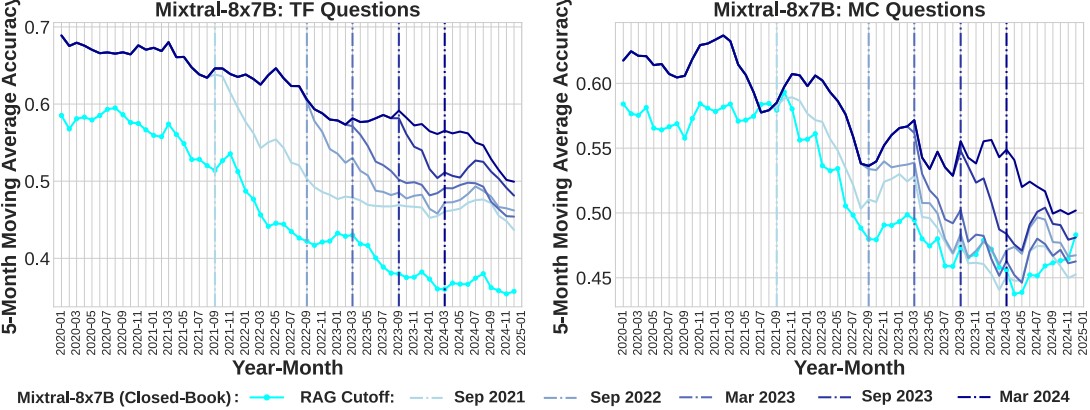

*Figure 15.* Results for Mixtral-8x7B in the constrained open-book setting.

# D. Prompts

All the prompts we use are shown in this section. The QA generation prompts and evaluation prompts are adapted from Zhang et al. (2024), and the prompt to categorize our generated questions is taken from Halawi et al. (2024).

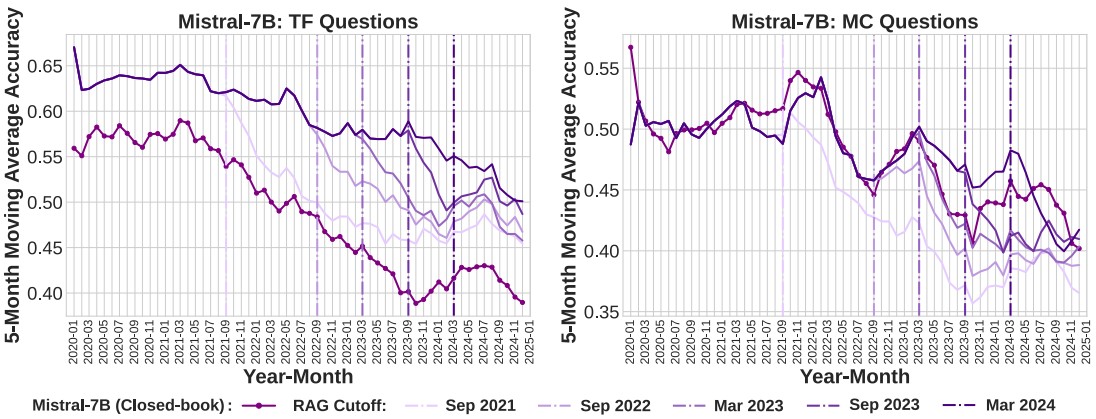

*Figure 16.* Results for Mistral-7B in the constrained open-book setting.

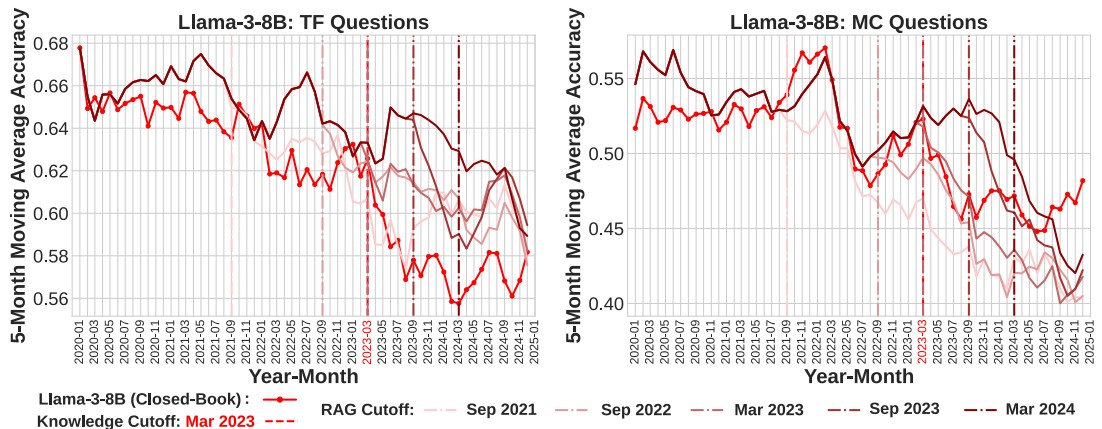

*Figure 17.* Results for Llama-3-8B in the constrained open-book setting.

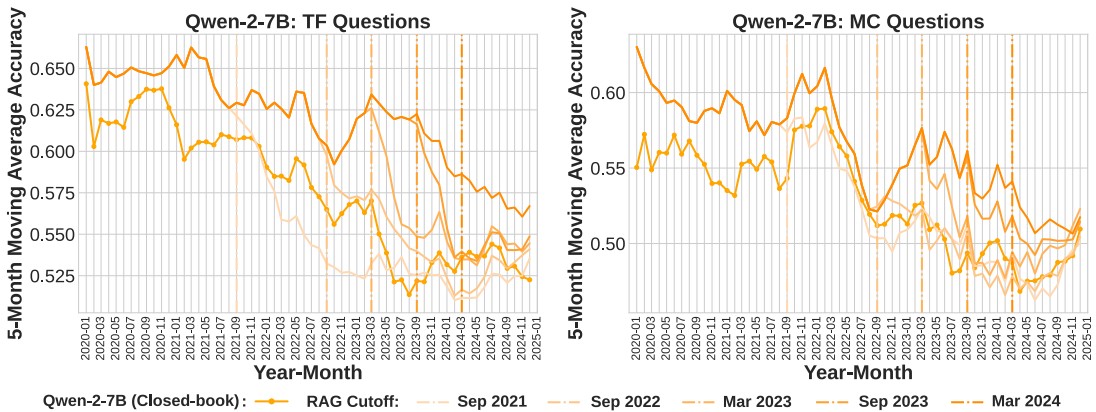

*Figure 18.* Results for Qwen-2-7B in the constrained open-book setting.

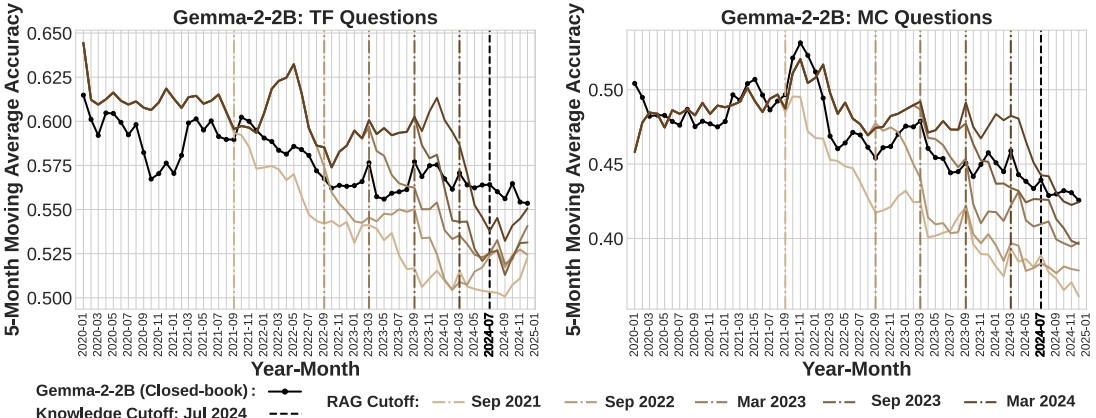

*Figure 19.* Results for Gemma-2-2B in the constrained open-book setting.

**[News Article]**
**Title:** DFL leaders, Minneapolis announce deal on rideshare pay, but Lyft and Uber say they will leave
**Publishing Date:** 2024-05-06
**Text:** ST. PAUL, Minn. — Key lawmakers and DFL legislative leaders on Monday announced that they have a compromise on statewide rideshare regulations in Minnesota, but Uber and Lyft quickly rejected the plan and said that they will still exit the market if the proposal becomes law.

The amendment, which will be heard in a House committee Tuesday, includes a minimum wage pay rate of $1.27 per mile and 49 cents per minute. Leaders said they worked with the Minneapolis City Council on finding compromise, but the companies were not included in the negotiations.

The proposal is lower than the Minneapolis rate of $1.40 per mile and $0.49 per minute, but higher than study done by the Minnesota Department of Labor and Industry, which determined $0.89 mile and $0.49 for the Twin Cities metro would bring driver pay up to the city's minimum wage of $15.57 an hour.

"This was a really big move for the Minneapolis City Council. This is coming down 13 cents from the rate that they passed. And this is consistent with the rates that Uber and Lyft are operating with and other states and other jurisdictions," House Majority Leader Jamie Long, DFL-Minneapolis, who was a key negotiator, told reporters. "We know that the companies have said the same thing in other states. They've said that they were going to leave at certain rates, and they've stayed and been able to operate profitably."

Uber and Lyft quickly responded that they will still plan to leave Minneapolis — as they said would happen on July 1, when the city's ordinance is effective — and the state if this becomes law.
...

---

**[Generated QA Pairs]**
**Question:** What will be the response from Uber and Lyft if the proposed rideshare regulations become law in Minnesota by May 2024?
**Answer:** (a) They will plan to leave the market.
**Misleading Choices:**
(b) They will stay and adjust their pricing to meet regulations.
(c) They will collaborate to develop an alternative service model
(d) They will reduce their services rather than exit.

---

**[Closed-Book Setting]**

• The question is asked directly to Mixtral-8x7B.

• Model response: **(wrong)** "(d) They will reduce their services rather than exit. This is because leaving the market entirely can be a drastic move and may not be the first response to new regulations. Reducing services is a more likely response, as it allows rideshare companies to comply with the new laws while still maintaining a presence in the market. Additionally, they may also try to negotiate or influence the regulations before they become law."

**[Constraint-Open Book Setting]**

• The RAG-cutoff refers to the latest accessible date for retrieving articles. With RAG-cutoff = 2024-03-01, we extracted the top 5 relevant articles (here, only the titles are displayed, but the articles are included in the evaluation prompt):

  – (2023-05-25) Uber Threatens To Cut Most Service In Minnesota If Minimum Pay Law Takes Effect
  – (2023-05-25) Minnesota Governor Vetoes Rideshare Minimum Pay Bill After Uber Threatens To Leave State
  – (2023-08-09) Minneapolis City Council moving towards minimum wage for rideshare drivers
  – (2024-02-27) Minneapolis City Council to consider rideshare minimum wage ordinance Tuesday
  – (2024-01-11) Some Lyft, Uber drivers to strike Thursday at MSP Airport

• Model response: **(correct)** "(a) They will plan to leave the market."

**[Gold Article Setting]**

• In the gold article setting, models are provided direct access to the gold article, from which the question is generated. The question then is treated as a reading comprehensive question rather than a forecasting question.

• Model response: **(correct)** "(a) They will plan to leave the market."

*Figure 20.* An example of evaluating Mixtral-8x7B across different settings.

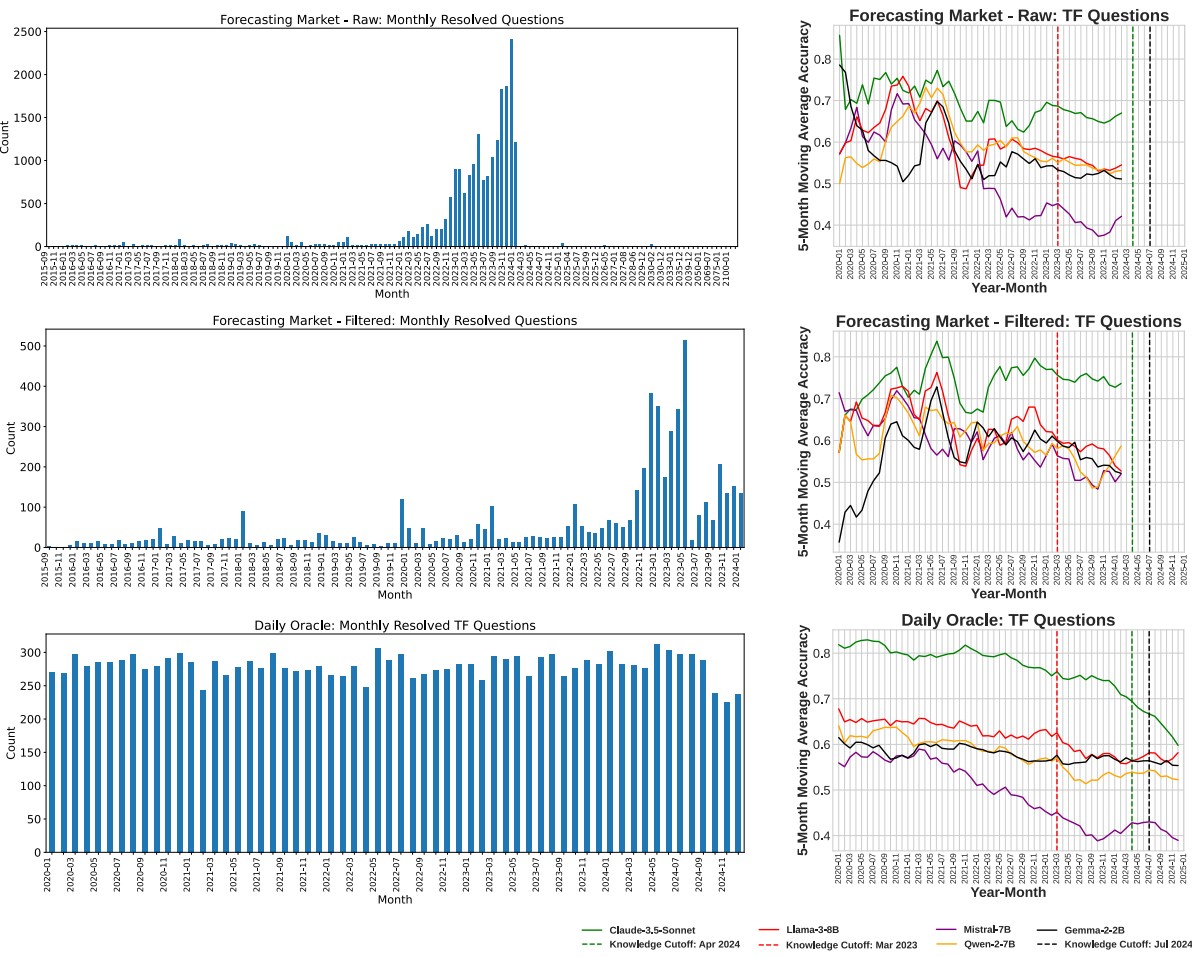

*Figure 21.* Dataset size and closed-book evaluation accuracy over time on the forecasting market dataset.

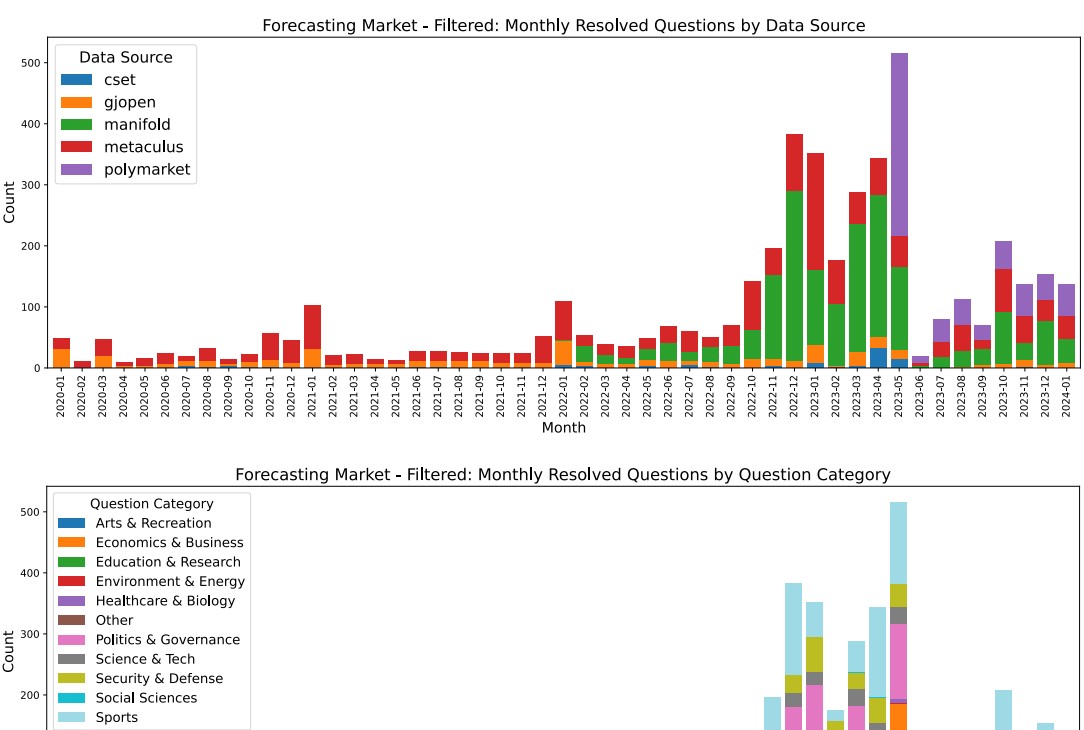

*Figure 22.* Distribution shift of the filtered forecasting market dataset.

You are an expert in extracting summary and keypoint from articles.

# Rules
1. Provide a comprehensive summary of the entire article in one paragraph, ensuring that all essential aspects are addressed. Your summary should include key statistics, notable dates, and any significant statements to fully convey the context and content of the news story.
2. Please provide one keypoint that summarizes the new event from the article with the following rules:
- Focus specifically on events that are newly occurring on the publication date of the article. If the article does not introduce a new event but instead discusses ongoing topics or is about non-news content like advertisements, state 'No new event reported.'
- The point should be concise, accurate and complete, especially for numbers, names and dates.
- Basically NO "he, she, they, it, them, etc" are allowed. Please clearly write out the entity you are referencing in the point.
- You are not allowed to start with any of the phrases: the article discusses, the article shows, the article emphasizes, the article discusses, the speaker says, the speaker discusses, the author mentions, etc.

# Examples
Here are several examples of extracting keypoints from articles. Note that the articles in different examples are irrelevant.
## Example 1:
Article: Professional golfer Lexi Thompson has announced her retirement from professional golf at the end of the 2024 season at the age of 29. Thompson, an 11-time LPGA Tour champion, made the announcement ahead of her 18th consecutive US Women's Open appearance. She turned professional in 2010 and won her first major at the 2014 Kraft Nabisco Championship. Despite enduring injuries that led to a drop in her world ranking, Thompson continued to compete at a high level. In her retirement announcement, Thompson expressed gratitude for the opportunities golf provided her and highlighted her excitement for the next chapter of her life.
Publishing Date: 2024-05-29, Wednesday
Keypoint: Lexi Thompson announced her retirement from professional golf at the end of the 2024 season at the age of 29.

## Example 2:
Article: India's capital territory of Delhi sweltered to its highest-ever temperature of 49.9 degrees Celsius (121.8 degrees Fahrenheit) on Tuesday, as an oppressive heat wave forced authorities to impose water rationing.

The Indian Meteorological Department (IMD) said the new record was measured in the suburb of Mungeshpur, surpassing Delhi's previous high of 49.2 degrees Celsius (120.5 degrees Fahrenheit), observed in May 2022.
Publishing Date: 2024-05-29, Wednesday
Keypoint: Delhi experienced its highest-ever temperature of 49.9 degrees Celsius on 2024-05-28, Tuesday.

# New Article
Based on the provided rules and examples, please summarize the article and identify one key point that concludes the new event on the publishing date.
Article: {}
Publishing Date: {}

# Output: Output should follow the format of
Summary:
Keypoint:

*Figure 23.* Prompt in the *Article Summary* step, adapted from Zhang et al. (2024).

[Setup:] Today is {}. You will be testing people who are from the past, i.e. a week or a month ago. A recently published article will serve as the basis for your questioning. Your objective is to ask them questions that assess the accuracy and plausibility of their predictions about events.

You will write question-answer pairs:
1. The question should challenge the person to predict an outcome or development related to the article's content as if they were being asked one week or one month ago. Please provide the question that can be answered on {}, but only guessable not answerable before {}.
2. The answer MUST be based on factual information from the article. Ensure that the answers do not predict outcomes that have not been explicitly stated in the article.

[Rules:]
Article: {}.
Publishing date: {}

Please generate four questions about the above article, along with answers. You should follow the instructions below:
1. Please turn the key point "{}" into the question, with focusing more on whether the event will happen.
2. The question should NOT be designed for reading comprehension. Please focus more on what happened rather than the implications after the event.
3. The question MUST be in future tense.
- Start the first question with "Will", with the answer as "Yes".
- Start the second question with "Will", with the answer as "No".
- Start the third and fourth questions with a phrase like "What will", "Who will", "Where will", "Which xxx will", "How much will", or "How many will".
4. There must be a time element in the question. It can be phrases like "In {} ...", "By {}, ...", "... in {}?".
5. You MUST NOT use unclear implicit time element phrases like "in the future" or "in the upcoming weeks".
6. You should avoid: questions that require numerical reasoning; questions that require substantial world knowledge.
7. The answer MUST be short and concise, avoiding using redundant words or repeating the information in the question.
8. The question must be grammatically correct and contain the information required to answer. NO "he, she, they, it, them, etc" allowed. Please clearly write out the entity you are referencing in the question.
9. The question MUST be able to be answered by the article.
10. The question MUST NOT include the information that came out just now. It should be understandable to people from the past. Avoid using "How will" or "Why will" questions, as they imply that the event has already occurred.

[Suggested questions and questions to avoid are detailed below:]

- Keypoint: Delhi experienced its highest-ever temperature of 49.9 degrees Celsius on Tuesday, leading to water rationing due to the oppressive heat wave.
- Suggested Question: Will Delhi break the highest temperature record again by May 2024?
- Avoid This Question: Will extreme heat events continue to pose a threat to India's development in the upcoming years?
- Reason to Avoid: The time constraint "in the upcoming years" is vague and the question can not be answered based on today's knowledge.

*Figure 24.* Prompt in the *QA Generation* step (part 1), adapted from Zhang et al. (2024).

- Keypoint: Owners of nearly 84,000 older Nissan vehicles in the United States equipped with recalled, unrepaired Takata air bags, including models such as the 2002-2006 Nissan Sentra, are advised by NHTSA to immediately stop driving them due to safety concerns.
- Suggested Question: Will the older Nissan vehicles such as the 2002-2006 Nissan Sentra exhibit quality issues by May 2024?
- Avoid This Question: Will owners of the 2002-2006 Nissan Sentra, 2002-2004 Nissan Pathfinder, and 2002-2003 Infiniti QX4 heed the NHTSA's advice to immediately stop driving their vehicles in late May 2024?
- Reason to Avoid: This question is overly specific. People from the past would not have known the "NHTSA's advice".

- Keypoint: Children's sketches of violent scenes, likely made by children aged 5-7 before the eruption of Mt. Vesuvius in 79 AD, have been uncovered at the archaeological park of Pompeii.
- Suggested Question: Will children's sketches of violent scenes be discovered at the archaeological park of Pompeii by May 2024?
- Avoid This Question: Will the newly discovered children's sketches at the archaeological park of Pompeii be available for public viewing by May 2024?
- Reason to Avoid: This question includes future events about newly discovered children's sketches in Pompeii, which wouldn't be known to a past audience.

- Keypoint: North Korea has been sending "filth and garbage" across the border to South Korea using giant balloons as a new strategy, prompting South Korean authorities to warn of the objects landing in residential areas. The move, according to North Korean state media KCNA, was to retaliate against South Korean activists who often send materials to the North.
- Suggested Question: What will North Korea do to retaliate against South Korean activists who often send materials to the North by May 2024?
- Avoid This Question: Will North Korea continue using balloons to send items across the border to South Korea by May 2024?
- Reason to Avoid: The word "continue" should not be used here. The question MUST NOT include the information that came out just now.

[Output:] Now please write four clear and concise question-answer pairs following the instructions and examples above. Once again the question should NOT be designed for reading comprehension but of forecasting interests. Also, vague and implicit time elements like "in the future", "in the upcoming weeks" or "in the coming years" should NOT be used. The question should be able to answer on {}, but only guessable not answerable before {}. You should output the question along with its answer, in the format of
'''
Question 1: "Will xxx?"
Answer 1: Yes.

Question 2: "Will xxx?"
Answer 2: No.

Question 3: Either "What will xxx?", "Who will xxx?", "'Where will xxx?", "Which xxx will", "How much will xxx?", or "How many will xxx?"
Answer 3: xxx.

Question 4: Either "What will xxx?", "Who will xxx?", "Where will xxx?", "Which xxx will", "How much will xxx?", or "How many will xxx?"
Answer 4: xxx.
'''

*Figure 25.* Prompt in the *QA Generation* step (part 2), adapted from Zhang et al. (2024).

# Rules
Article: {}
Given the article, please generate three noising answers to the given questions, whose correct answers can be obtained from the article. Name the three noising answers as (b), (c) and (d) respectively. While (b), (c) and (d) should all be unambiguously incorrect, they should also make sense and be plausible.

# Examples
Here are examples showing the output format. This example is NOT related to the noising answers you will generate.

Question: What will be the annual change in the UK's Consumer Prices Index (CPI) for November 2021?
Correct Answer: 'Less than 1.7%'
Noising Answers:
(b) 'Between 1.7% and 2.2%, inclusive'
(c) 'More than 2.2% but less than 2.9%'
(d) '2.9% or more'

Question: Who will win the 2020 Georgia Democratic primary?
Correct Answer: 'Joe Biden'
Noising Answers:
(b) 'Michael Bloomberg'
(c) 'Pete Buttigieg'
(d) 'Someone else'

Question: Before July 2020, will it be officially announced that the Tokyo 2020 Summer Olympics and/or Paralympics will be postponed, canceled, and/or relocated?
Correct Answer: Yes, the Olympic Games only
Noising Answers:
(b) 'Yes, the Paralympic Games only'
(c) 'Yes, both'
(d) 'No'

# Input:
Question 1: {}
Correct Answer 1: {}

Question 2: {}
Correct Answer 2: {}

# Output: Now please generate three noising answers to the question, given the above article, instructions and examples. DO NOT output the backgrounds, the question or any other explanations.
Noising Answers 1:
(b) xxx.
(c) xxx.
(d) xxx.

Noising Answers 2:
(b) xxx.
(c) xxx.
(d) xxx.

*Figure 26.* Prompt in the *Misleading Choices Generation* step, adapted from Zhang et al. (2024).

# Task
Please help evaluate the quality of question-answer pairs derived from the given news article. The questions will be presented to someone who has not seen the corresponding news article, in order to evaluate the accuracy and plausibility of the event prediction ability.

# Inputs
Article: {}
Publishing Date: {}
Question 1: {}
Answer 1: {}
Question 2: {}
Answer 2: {}
Question 3: {}
Answer 3: {}
Question 4: {}
Answer 4: {}

# Scoring Categories
## Correctness: Given the above article, please check if the answer is correct to the question with 100% certainty.
- 2 points: There is evidence in the article that the answer is correct with 100% certainty.
- 1 point: The answer generally aligns with the news facts but has minor inaccuracies or missing details.
- 0 point: Significantly misaligned with the news facts.

## Only Answerable on Publishing Date: Imagine traveling back in time to one week before the article's publishing date ({}). At that time, you are asked the question without having seen this specific article, but you do have access to all earlier news articles. The question should ideally be only guessable—not definitively answerable—based on the information available at that time. That is, the answer should be able to be found in the given article, but it should not be obtainable from earlier articles. Note that past tense descriptions in the article DO NOT INFLUENCE this assessment.
- 2 points: The question is answerable on {}, but only guessable not answerable before {}.
- 1 point: Could be somewhat predicted before {}, but not with complete certainty.
- 0 point: A person (could be anyone, even an expert in the field) would be able to find an article (or many) published before {} that answers the question with 100% certainty.

### 0 point examples
Example 1:
Question: What will be one of Lexi Thompson's career highlights in professional golf?
Answer: Winning 11 LPGA Tour titles.
Reasoning: This question is answerable with prior knowledge and does not test predictive ability related to the publishing date.

## No New Information: Ensure the question does not include new information that only became known on the publishing date, making it understandable for a past audience.
- 2 points: No new information from the publishing date are included.
- 1 point: Minor new information from the publishing date might be inferred but are not explicitly stated.
- 0 point: Includes clear new information from the publishing date, unsuitable for past understanding.

### 0 point examples
Example 1:
Question: Will owners of the 2002-2006 Nissan Sentra, 2002-2004 Nissan Pathfinder, and 2002-2003 Infiniti QX4 heed the NHTSA's advice to immediately stop driving their vehicles in late May 2024?
Reasoning: This question contains new information on the publishing date. People from the past would not have known the "NHTSA's advice".

*Figure 27.* Prompt in the *QA Filtering* step (part 1).

Example 2:
Question: "What will Lexi Thompson's ranking be at the time of her retirement announcement in May 2024?"
Reasoning: This question contains the information that Lexi will annouce her retirement, which is not known to the people from the past.

Example 3:
Question: "Will the newly discovered children's sketches at the archaeological park of Pompeii be available for public viewing by May 2024?"
Reasoning: This question includes future events about newly discovered children's sketches in Pompeii, which wouldn't be known to a past audience

## Objectiveness: The answer should not rely more on the author's personal views than on objective facts.
- 2 points: Completely objective, based strictly on reported facts.
- 1 point: Primarily objective, with minor subjective interpretations.
- 0 point: Largely subjective or opinion-based, lacking a factual basis.

## Clear Time Element: This category checks if the question has a clear element in it, without having vague phrases like "in the future" or "in the upcoming weeks".
- 2 points: The question has clear time elements, like "by May 2024" or "in July 2023".
- 1 point: The question includes a general timeframe, like "next month" or "this winter," which allows for some estimation but lacks precise dates.
- 0 point: The question includes vague time phrases like "in the future" or "in the upcoming weeks," which do not specify a clear or precise timeframe.

### 0 point examples
Example 1:
Question: Will extreme heat events continue to pose a threat to India's development in the upcoming years?
Reasoning: The time constraint "in the upcoming years" is vague.

Example 2:
Question: "What will Illinois require from parents who monetize their children's online activities starting in July?"
Reasoning: The mention of "July" specifies only the month and lacks the necessary detail of the year.

## Public Interest: Determine if the question addresses a topic of public concern.
- 2 points: The question covers a topic that widely affects or interests the public.
- 1 point: The question is of moderate interest, relevant to specific groups.
- 0 point: The topic is overly personal or localized, lacking relevance to the broader public.

### 0 point examples
Example 1:
Question: Will the exhibition 'Fragile Beauty' at London's Victoria & Albert Museum include both midcentury and contemporary works in May 2024?
Reasoning: The specific details of an personal art exhibition's contents are generally of limited public interest.

## Answer Not Too Obvious: This category evaluates whether the answer to a question is too predictable or straightforward based on the question itself.
- 2 points: The answer provides new or non-obvious insights, requiring additional context or understanding not explicit in the question.
- 1 point: The answer is somewhat predictable but includes minor additional information or a slight twist.
- 0 point: The answer directly restates or closely mirrors the question, offering no new details or insights.

*Figure 28.* Prompt in the *QA Filtering* step (part 2).

### 0 point examples
Example 1:
Question: What will New York officials do to ensure safety for the ICC Men's T20 Cricket World Cup following global threats from ISIS-K?
Answer: New York officials will implement increased safety precautions for the event.
Reasoning: The answer is straightforward and expected, as it directly restates the premise of the question without providing any new or specific details on how the safety precautions will be implemented or what they might entail.

# Instructions
Evaluate each question-answer pair by assigning points in each of the categories based on the criteria provided. Please be strict on giving points. If the requirements of a category are not fulfilled, assign a point of 0.

# Please strictly follow this output template:
*Question 1*
## Correctness
- Reasoning:
- Point:
## Only Answerable on Publishing Date
- Reasoning:
- Point:
## No New Information
- Reasoning:
- Point:
## Objectiveness
- Reasoning:
- Point:
## Clear Time Element
- Reasoning:
- Point:
## Public Interest
- Reasoning:
- Point:
## Answer Not Too Obvious
- Reasoning:
- Point:

*Question 2*
## Correctness
- Reasoning:
- Point: ## Only Answerable on Publishing Date
- Reasoning:
- Point:
## No New Information
- Reasoning: - Point:
## Objectiveness
- Reasoning:
- Point:
## Clear Time Element
- Reasoning:
- Point:

*Figure 29.* Prompt in the *QA Filtering* step (part 3).

```
## Public Interest
- Reasoning:
- Point:
## Answer Not Too Obvious
- Reasoning:
- Point:

*Question 3*
## Correctness
- Reasoning:
- Point:
## Only Answerable on Publishing Date
- Reasoning:
- Point:
## No New Information
- Reasoning:
- Point:
## Objectiveness
- Reasoning:
- Point:
## Clear Time Element
- Reasoning:
- Point:
## Public Interest
- Reasoning:
- Point:
## Answer Not Too Obvious
- Reasoning:
- Point:

*Question 4*
## Correctness
- Reasoning:
- Point:
## Only Answerable on Publishing Date
- Reasoning:
- Point:
## No New Information
- Reasoning:
- Point:
## Objectiveness
- Reasoning:
- Point:
## Clear Time Element
- Reasoning:
- Point:
## Public Interest
- Reasoning:
- Point:
## Answer Not Too Obvious
- Reasoning:
- Point:
```

*Figure 30.* Prompt in the *QA Filtering* step (part 4).

**System Prompt:** You're an expert in forecasting events. You will NEVER refuse to answer a forecasting question by saying "I cannot predict the future", even if without 100% certainty.

**User Prompt:** You should output your answer as either 'Yes' or 'No' WITHOUT anything else.

Question: {}
Choices: 'Yes' or 'No'
[Output:] Your answer:

*Figure 31.* Closed-book evaluation prompt for TF questions, adapted from Zhang et al. (2024).

**System Prompt:** You're an expert in forecasting events. You will NEVER refuse to answer a forecasting question by saying "I cannot predict the future", even if without 100% certainty.

**User Prompt:** You should output your answer as either '(a)', '(b)', '(c)' or '(d)' WITHOUT anything else.

Question: {}
Choices:
(a) {}
(b) {}
(c) {}
(d) {}
[Output:] Your answer:

*Figure 32.* Closed-book evaluation prompt for MC questions, adapted from Zhang et al. (2024).

**System Prompt:** You're an expert in forecasting events. You will NEVER refuse to answer a forecasting question by saying "I cannot predict the future", even if without 100% certainty.

**User Prompt:** You should output your answer as either 'Yes' or 'No' WITHOUT anything else. Below are the top 5 relevant news article fragments retrieved for the question, which may or may not assist you in making a forecast.
Article 1: {}
Article 2: {}
Article 3: {}
Article 4: {}
Article 5: {}

Question: {}
Choices: 'Yes' or 'No'
[Output:] Your answer:

*Figure 33.* Constrained open-book evaluation prompt for TF questions, adapted from Zhang et al. (2024).

---

**System Prompt:** You're an expert in forecasting events. You will NEVER refuse to answer a forecasting question by saying "I cannot predict the future", even if without 100% certainty.

**User Prompt:** You should output your answer as either '(a)', '(b)', '(c)' or '(d)' WITHOUT anything else. Below are the top 5 relevant news article fragments retrieved for the question, which may or may not assist you in making a forecast.
Article 1: {}
Article 2: {}
Article 3: {}
Article 4: {}
Article 5: {}

Question: {}
Choices:
(a) {}
(b) {}
(c) {}
(d) {}
[Output:] Your answer:

---

*Figure 34.* Constrained open-book evaluation prompt for MC questions, adapted from Zhang et al. (2024).

---

**System Prompt:** You're an expert in forecasting events. You will NEVER refuse to answer a forecasting question by saying "I cannot predict the future", even if without 100% certainty.

**User Prompt:** You should output your answer as either 'Yes' or 'No' WITHOUT anything else. Below is the updated news article relevant to the question, which may help you in providing an answer.
Article: {}

Question: {}
Choices: 'Yes' or 'No'
[Output:] Your answer:

---

*Figure 35.* Gold article evaluation prompt for TF questions, adapted from Zhang et al. (2024).

---

**System Prompt:** You're an expert in forecasting events. You will NEVER refuse to answer a forecasting question by saying "I cannot predict the future", even if without 100% certainty.

**User Prompt:** You should output your answer as either '(a)', '(b)', '(c)' or '(d)' WITHOUT anything else. Below is the updated news article relevant to the question, which may help you in providing an answer.
Article: {}

Question: {}
Choices:
(a) {}
(b) {}
(c) {}
(d) {}
[Output:] Your answer:

---

*Figure 36.* Gold article evaluation prompt for MC questions, adapted from Zhang et al. (2024).

Question: {}
Options:
- Science & Tech
- Healthcare & Biology
- Economics & Business
- Environment & Energy
- Politics & Governance
- Education & Research
- Arts & Recreation
- Security & Defense
- Social Sciences
- Sports
- Other
Instruction: Assign a category for the given question.
Rules:
1. Make sure you only return one of the options from the option list.
2. Only output the category, and do not output any other words in your response.
3. You have to pick a string from the above categories.
Answer:

*Figure 37.* Prompt to categorize the generated questions, taken from Halawi et al. (2024).

