# OpenReview forum: "Are LLMs Prescient? A Continuous Evaluation using Daily News as the Oracle"
_ICML.cc/2025/Conference — ICML 2025 poster_

### Official Review · Reviewer_dDkL · 2025-03-15

**Overall Recommendation:** 3

**Summary:**

The manuscript introduces Daily Oracle, a benchmark dataset composed of automatically-generated question-answer pairs concerning daily news over a 4 year period. The questions are all phrased in a "forecast" manner (e.g., "Will X happen?", "What will Y be on DD-MM-YY?") and are either yes/no or multiple (4) choices questions.

Different language models are evaluated on this benchmark in one of three regimes: closed-book (no context), constrained open-book (RAG on news articles up to a certain cutoff date), and gold article (the article used to generate the question-answer pair is explicitly provided to the model). The models are good at predicting the answers in the gold article setting. In the two other settings, the models are better at predicting the sought answer at early time than at latter ones, with more pronounced decrease around the data cutoff point. Part of this decrease has been identified as the model refusing to predict the future.

**Claims And Evidence:**

### Claim 1: Introducing a continuous forecasting evaluation benchmark

> We present Daily Oracle, the largest and most up-to-date forecasting dataset

From Table 1, this dataset is the only daily one, it is one of 6 forecasting datasets (which, as I understand it, means that the questions are worded as a future-prediction task), it is the second largest (and the largest dates back to 2021). I deem this claim supported.


### Claim 2: Empirical Findings on Performance Degradation

> Our work effectively reveals a clear performance degradation pattern in LLMs’ forecasting accuracy over time.

The statement as worded appears factual. However, I see issues with the general narrative, as well as with more granular claims made thorough the manuscript. This is my main issue with the manuscript in its current state, which I try to convey through comments on some excerpts (emphasis mine).

> Additionally, despite prompting the models to avoid **responses like “I cannot predict the future”** and instead provide definitive answers, there are cases where such refusals still occur. The rejection rates are provided in the Appendix B.3, and **these cases are counted as incorrect** to ensure comparability across model results.

These models were explicitly trained to say things like “I cannot predict the future” when asked to predict the future. The refusal rates shown in Figure 8 are quite substantial, and they behave like one would expect them to for models that were trained to refuse to predict the future.

> However, post-knowledge cutoff, we observe steeper declines in many models, with GPT-4 showing the most drastic drop in MC performance, declining by 18.54%, compared to just 4.23% before the cutoff. This contrast highlights that while LLMs manage to retain a baseline of past knowledge with small degradation, **their ability to forecast future events deteriorates much more rapidly as they move beyond their training data, struggling with temporal generalization.**

... or they just do as they were trained to do? We're talking about more than 50% variation for Mixtral-8x7B on MC questions, and about 15% variation for Mistral/Mixtral on TF. If you add these values to those in Figure 3 (i.e., if you reinterpret refusals as correct answers), the corresponding TF curves would be flat, and Mixtral's MC curve would go up. And Figure 8 shows "lower bound" effect, only reporting the refusals that were caught, and not accounting for the cognitive dissonance induced by a prompt saying the opposite of what the model was trained to do.

> For Mixtral-8x7B, as the RAG cutoff dates extend to closer to the resolution dates, we observe a clear improvement in performance, indicating the model benefits from increasingly updated information retrieval. However, there are noticeable performance drops immediately after each RAG cutoff date when compared to providing information up to the day before the resolution date. **This highlights the importance of keeping up-to-date information for optimal RAG performance.**

No! This highlights that Mixtral refuses to answer when it doesn't have the information required to answer!

> The overall decline trend may come from two sources, the missing knowledge of future and a lack of up-to-date language representation.

Or these models were not trained to be oracles? I mean, even rewording the questions to become "which is the most likely..." assessments instead of "will ..." could potentially help. A more passive approach to study this phenomenon could be to assess if there is a difference in the wording of the questions that Mixtral *did* answer past the cutoff. Another data point: Claude is the least reluctant to predict the future (Figure 8) and the "best" at doing so (Figure 3).

To be clear, I do believe that there is something worth saying here, it is just that it may not be what is currently being said. This is the topic of my Question 1 below.

**Essential References Not Discussed:**

.

**Experimental Designs Or Analyses:**

See my answer to Methods and Evaluation Criteria above.

**Methods And Evaluation Criteria:**

Excluding the subject of my Question 1 below, the way the language models are evaluated makes sense to me. The reported metrics in the main paper are all averaged over 5 months, but the Appendix provides some raw data. This was important for me to make my mind about potential issues with the September 2024 transition to GPT-4o for generating the dataset.

The manuscript's main point is to introduce a benchmark, so the real evaluation concerns the benchmark itself. Summary statistics are provided in Figure 1, and distribution stability is assessed through Figure 2. Human evaluation according to different criteria is provided in A.4. Only 60 questions where human-annotated, there were only 4 human annotators, and there is no mention of their sociocultural background nor of their potential affiliation/overlap with the authors.

The generation method of the benchmark makes sense to me overall, except that there is no explicit check to assess whether the switch to GPT-4o in September 2024 altered the distribution of questions and answers.

**Other Comments Or Suggestions:**

Table 1: consider adding `\citep{...}` after the dataset names. As a reader, I often use those as lookup tables.

The ordinate axes in Figure 8 appear mislabeled.

Many news articles may be generated by LLMs toward the end of the period, whereas this likely wasn't as much the case back in 2020.

When the model data cutoff date is unknown, consider showing the model's release date: the cutoff must precede it (with caveats for API-based models).

Consider giving high-level description of the annotators. Things like their level of education, age range, etc., as well as their relations to the authors, if any, and if they were they paid.

**Other Strengths And Weaknesses:**

The benchmarking dataset itself is likely to be useful to the community.

**Questions For Authors:**

### Question 1: What is your proposed solution to my issues mentioned in Claim 2 above?

Do you agree with my observations and assessments? Are you open to contextualize, reword, and/or tone down these kind of claims? If yes, to what exactly?

Or if you disagree, please let me know of your arguments and/or additional evidence.

**Relation To Broader Scientific Literature:**

What is presented in the Related Work section makes sense to me, though I may be unaware of some missing related work.

**Theoretical Claims:**

I didn't notice any particular theoretical claims.

---

> ### Author Rebuttal · Authors · 2025-04-01
>
> We sincerely thank the reviewer for the thoughtful feedback and constructive suggestions, which we will incorporate into the future version of our manuscript.
> ***
> ### Concerns about refusal rate
> We appreciate the reviewer’s thoughtful comments regarding the refusal cases of Mistral and Mixtral models. We would like to clarify the following points: as seen in Figure 8, **only the Mistral and Mixtral models exhibit notable refusal rates**, with approximately 10–30% on TF questions and 1.5–8% on MC questions. In comparison, Qwen-2-7B and Gemma-2-2B show relatively low refusal rates—<5% for TF and <2% for MC—while **all other models have near-zero refusal rates for TF and <1% for MC**. We count refusal cases as incorrect both to maintain comparability across models and because failing to provide an answer—when a prediction is expected—represents an unsatisfactory outcome from the user's perspective.
>
> Further,
> 1. We provide an additional plot [Fig.S7](https://imgur.com/a/ZTpSKyj) that excludes questions the models refused to answer, focusing only on cases where the model provided a definite answer. While this means the results among models are no longer directly comparable, the performance degradation trend remains evident, particularly for TF questions in Mixtral-8x7B. We hope this analysis helps address the reviewer’s concern regarding the impact of refusal behavior on the observed trends. We would be happy to include this plot in a future version of the paper to help clarify concerns related to refusal cases.
>
> 2. Our claim of `"their ability to forecast future events deteriorates much more rapidly as they move beyond their training data, struggling with temporal generalization"` specifically contrasts pre- and post-knowledge-cutoff performance for models with known cutoff dates. The Mistral and Mixtral models mentioned by the reviewer do not disclose such information, and therefore are not used to support the claim here. For the models with known knowledge cutoff (Claude, GPT, LLaMA, and Gemma), refusal rates are minimal—most of them having nearly 0% for TF questions and less than 1% for MC questions—and thus have negligible impact on the observed degradation patterns.
>
> 3. **Effect of rewording questions:** We thank the reviewer for suggesting the rewording experiment, and tested the impact of softer phrasing (e.g., rephrasing “Will...” to “Would it be likely...”) using Qwen-2-7B and Mistral-7B on TF questions. As shown in [Fig.S8](https://imgur.com/a/NMy925N), this change reduced refusal rates by approximately 10%. However, the overall performance degradation trend still persists, suggesting that refusal alone does not account for the observed decline.
>
> 4. **On models `“not being trained to be oracles”`:** We believe that model refusals alone do not explain the observed performance decline. As shown in Fig.S7, even when refusal cases are excluded and only answered questions are considered, the downward trend in performance persists.
>
> 5. **On reluctance vs. performance:** While Claude demonstrates the lowest refusal rate on TF questions (Figure 8), and high performance (Figure 3), this pattern does not generalize across models. GPT-3.5, GPT-4, and LLaMA-3-8B also have near-zero refusal rates, yet show varying levels of forecasting accuracy. Thus, we do not find strong evidence that lower reluctance to predict directly correlates with better performance.
>
> Overall, our findings indicate that performance decline is a general phenomenon that persists even when refusals are removed from the analysis.
> ***
> ### Others
> 1. For concerns about the number of human evaluation samples, please refer to our response to Reviewer xifq ("Sample size of human evaluation"). Regarding the shift of newer 4o models since September 2024, please refer to our response to Reviewer xifq ("Distribution of question categories and question types") and Reviewer s4SM ("Choice of models in QA generation").
> 2. Background of human annotators: The 4 annotators are graduate students from the authors’ institution, majoring in finance, accounting, statistics, and data science, respectively. They were not involved in the research beyond their role in conducting the evaluations.

---

> > ### Comment · Reviewer_dDkL · 2025-04-03
> >
> > Thank you for the clarification.
> >
> > > 3. Effect of rewording questions:
> >
> > A 5-10% decrease in refusal rate, yielding a ~5% increase in performances (even for Qwen), is significant. If I were reviewing manuscript introducing a modeling technique granting a 5% improvement across many architectures for such a use case, I would likely accept it.
> >
> > > which we will incorporate into the future version of our manuscript.
> >
> > Will this incorporation include the aforementioned dependency on the formulation of the question, and more generally the potential issues with using as oracles LLMs that were explicitly trained against speculation? I think that this should be discussed in introduction, perhaps even the abstract. If you are ready to discuss such points, could you please give some examples of sentences/paragraphs as to how you intend to present them?
> >
> >
> > ## April 7th Addendum
> >
> > (The interface does not allow me to reply to https://openreview.net/forum?id=v2nV83Q849&noteId=x87Dt626c3 , so I'm doing so here.)
> >
> > > Planned revisions for future manuscript
> >
> > I assess that such changes would have the manuscript cross the minimal threshold to come on the good side of the "truthful-misleading axis".
> >
> >
> > > interesting but outside the scope of our core research question
> >
> > I acknowledge that this was outside the initial scope. My point is that this work may have revealed a *fundamental flaw at the core of this "future event prediction" research space*, and that future work should properly ponder these questions at an early stage of experimental design. I believe that there may be a missed opportunity to clearly spread that message.
> >
> >
> > I am still not as satisfied as I would have wished to be, but I will raise my score from 2 to 3. I won't fight for nor against this manuscript.

---

> > > ### Author Response · Authors · 2025-04-06
> > >
> > > Thank you for the following comments. We hope to make further clarifications.
> > > ### 1. Effect of rewording questions is interesting but outside the scope of our core research question
> > > It is important to note that changing the prompting “Will” to “Would it be likely...” again provides a clear degrading trend and has a similar shape as the original prompts, thus the main claim still holds. While the observed improvement in performance is interesting and worth exploring, analyzing such effects of different prompting styles falls outside the scope of this work. Moreover, it is standard practice to frame forecasting questions using the “Will…” format rather than softer or speculative phrasing - consistently used across all forecasting datasets (ForecastQA, AutoCast, TLB-Forecast, ForecastBench, and FreshBench) mentioned in our literature review.
> > > ### 2. Refusal to answer as an indicator of lack of knowledge
> > > We acknowledge the reviewer's concern that models are designed to be cautious—sometimes refraining from answering questions about uncertain future events. This cautious behavior, particularly evident in Mistral and Mixtral (Figure 8), likely stems from their RLHF training, which discourages the output of potentially misleading information. For our evaluation, we count refusal cases as incorrect both to ensure comparability across models and because a failure to provide a prediction is unsatisfactory from a user’s perspective.
> > >
> > > In a closed-book setting, refusal rates of Mistral and Mixtral range from ~10% to 30%. However, when models are supplied with retrieved articles or gold articles (thus receiving additional relevant information), they are more likely to generate definitive answers, reducing the refusal rate, i.e. the refusal rates in [Fig.S12](https://imgur.com/a/jvJ6mUN) (d) and (f) are much less than in the closed-book setting (b). Therefore, refusal to answer can be partially mitigated by providing more relevant knowledge.
> > > ### 3. Our conclusion of degradation is still valid given the refusal behavior
> > > We argue that our main conclusion - “we can observe the performance degradation of LLMs in future event forecasting tasks in multiple experimental setups” - still holds, regardless of the model’s refusal behavior. As evidenced in [Fig.S12](https://imgur.com/a/jvJ6mUN) (a), (c), (e), which excludes all questions that the model refuses to answer, we can still observe a clear degradation trend across 3 settings. Therefore, one of the reasons for the low performance of Mistral and Mixtral is the refusal behavior, however, the decline is likely from the lack of future knowledge and out-of-date representations.
> > >
> > > Moreover, the increasing trend in the refusal rate further supports our claim that over time model performance degrades (models become increasingly unwilling to forecast). In contrast, the lower refusal rates in the RAG and gold article settings indicate that when models are provided with more recent, relevant knowledge, their performance improves. This finding again underscores the necessity of continual pretraining or supplying updated knowledge.
> > > ### 4. Minimal refusal behavior in most of the models
> > > As mentioned in the first paragraph in previous response, we argue that the refusal behavior does not affect the majority of models evaluated, nor does it undermine our main findings.
> > > ### 5. Planned revisions for future manuscript
> > > We thank the reviewer for highlighting refusal behavior and will clarify our claim and add a paragraph to the discussion (in italic).
> > >
> > > Claim: The overall decline trend may come from two sources, the missing knowledge of future and a lack of up-to-date language representation. *The absence of relevant future information can lead to two outcomes: either the model makes uninformed or incorrect predictions, or, in some cases, more likely to refuse to answer altogether. We observe this latter behavior notably in Mistral-7B and Mixtral-8x7B, where refusal rates are significantly higher compared to other models.*
> > >
> > > Paragraph: *While most models show minimal refusal behavior, Mistral-7B and Mixtral-8x7B frequently refuse to answer forecasting questions (Figure 8). This is likely influenced by alignment techniques, which discourage speculative or uncertain responses in the post-training stage. Although refusal rates contribute to lower scores for certain models, our results show that performance degradation trends persist even when refusals are excluded ([Fig.S12](https://imgur.com/a/jvJ6mUN) (a), (c), (e)). We consider refusal to answer an indicator of performance limitations in forecasting tasks, as it reflects the model’s lack of actionable knowledge. When models are supplied with more up-to-date and relevant information, their refusal rates decrease ([Fig.S12](https://imgur.com/a/jvJ6mUN) (d), (f)). This suggests that refusal is one example of the broader challenge of temporal generalization and reinforces the need for continual model updates or improved external knowledge integration.*

---

### Official Review · Reviewer_s4SM · 2025-03-15

**Overall Recommendation:** 4

**Summary:**

The paper uses the task of forecasting real-world events to demonstrate that LLM knowledge deteriorates on more recent questions, and this trend also holds for retrieval. It generates these forecasting questions between January 2020 and December 2024 using LLMs, sourcing information from news articles.

**Claims And Evidence:**

Yes, the claims made in the paper are focused, and supported with evidence. However, there are concerns with the experiment design, stated below.

**Essential References Not Discussed:**

Consistency Checks for Language Model Forecasters, Daniel Paleka, Abhimanyu Pallavi Sudhir, Alejandro Alvarez, Vineeth Bhat, Adam Shen, Evan Wang, Florian Tramèr, ICLR 2025 -- Also uses a sophisticated LLM generated forecasting questions pipeline. Please compare with this.

**Experimental Designs Or Analyses:**

The broad experimental design of varying model performance across time (including RAG cutoffs) makes sense and is interesting. However there are a few major issues in the details which could be confounders for the results:

1) The performance with RAG greatly depends on the number of recent articles in the index, which is being varied across time. Could you please present temporal plots with the number of articles published in a fixed time window, say the last year from each date, that are retrievable from the index?

2) Back-generating questions using news might lead to the creation of a biased question set based on events that are reported in the news. Why not use questions on platforms like Polymarket for this? Can you provide an empirical comparison to what happens if the same analysis was done using questions obtained from Polymarket's API?

**Methods And Evaluation Criteria:**

Please report brier score / log-odds for all plots. Accuracy is a bad metric for forecasting as one can realistically never be sure whether the event will occur or not (inherent uncertainty).

The key uncertainty about this paper lies in the quality of the LLM generated forecasting questions dataset. The paper does not discuss how it evaluates this quality. Further, GPT-3.5 is used to create some questions, and it is unclear why, when GPT4o/4omini are both cheaper and much more reliable models. I encourage the authors to think carefully about how the quality of questions can be measured, and also re-generate questions before September 2024 with GPT4o if this improves quality compared to GPT-3.5. This would make the data/benchmark much more usable for future work. For example, a quick look at the data provided in the supplementary led me to find obviously faulty questions. Eg:

a) "What career strategy will be recommended in January 2020, traditional ladder climbing or pursuing innovative and entrepreneurial approaches?" -- recommended by whom? I don't think this question has any single answer.

b) Which aspect of social media strategy will most brands and individual thought leaders overlook in January 2020? - Same issue as a).

One way of measuring question quality could be checking the brier score of the same models on questions beyond the cutoff-date, comparing different sources like Metaculus, Manifold, Polymarket as baselines. In fact, I am unsure why these other data sources are not already included in the analysis of this paper, as they do provide many thousands of questions and its unclear whether LLM generated questions provide any better quality.

**Other Comments Or Suggestions:**

Please add methodology to measure question quality to validate design choices, and address questions below.

**Other Strengths And Weaknesses:**

**Strengths**

1. The use of forecasting as a task to test model performance across time is interesting.

2. Generating forecasting questions using News and LLMs is a clever insight.

**Weaknesses**:
1. Question quality is not evaluated, and if the data is bad all the results could be unreliable.
2. The results/trends are confounded by the number of fresh articles in the retrieval index across time, which is not reported.

**Questions For Authors:**

1. Why is model performance increasing after July 2024 in the gold article setting?

2. Why use Mixtral 7b for Figure 4 (left) when its performance is almost worse than random after a certain point. Why not report the same Llama 3 8b model.

3. Could you add more details about the retrieval, such as the number of articles available at each month in the relevant period using your scraping pipeline? Would more sophisticated retrieval than BM25 lead to improved results for models?

4. Does question quality improve by using a better model?

**Relation To Broader Scientific Literature:**

There have been previous papers on the potential of forecasting as a language model benchmark. Backtesting and lookahead bias (a form of contamination) are extremely important for forecasting in other domains (such as stock market). This paper contributes towards better backtesting for language model forecasting.

**Theoretical Claims:**

No theoretical claims.

---

> ### Author Rebuttal · Authors · 2025-04-01
>
> We sincerely thank the reviewer for the detailed feedback.
> ***
> ### Brier score
> While we agree that Brier score is valuable to account for uncertainty in binary predictions, we clarify that accuracy remains a valid metric in this setting, revealing a clear performance degradation trend in our experiments. We provide Brier score result in [Fig.S4](https://imgur.com/a/fjtEONv). The increasing trend further confirms the previously identified trend of performance degradation. We also note that Brier score can be sensitive to calibration issues of LLMs, which is an important but orthogonal direction that we leave for future work.
> ***
> ### Quality of dataset
> We refer the reviewer to Section 3.1 & Appendix A.3 for our quality control process. During the QA filtering step, 7 principles are identified based on common mistakes observed during manual reviews while testing various QA generation prompts, and overall 18.11% TF questions and 24.20% MC questions were filtered out. Further validation, presented in Section 3.3 and Appendix A.4, includes a human evaluation of the filtering process. We find strong agreement between human reviewers and LLM-assigned scores, with an average accuracy of 89.52% across the 7 principles.
>
> We are aware of examples like those the reviewer mentioned and designed our filtering criteria such as “Answerability” to address such issues. While some imperfections are inevitable in any LLM-generated dataset, we note that for final QA pair acceptance, the agreement between LLM and human evaluations achieved 85% accuracy, indicating that the majority of the retained questions are valid and of acceptable quality. Moreover, if the questions were broadly unanswerable, we would not observe clear differences in LLM accuracy across models or the consistent degradation trends over time.
> ***
> ### Comparison with forecasting markets
> We would like to explain why we chose to focus on LLM-generated questions in this work:
> 1. [Prior work](https://arxiv.org/abs/2402.18563) collected data from 5 existing forecasting platforms, sourcing 48,754 raw questions (2015-2024). They note that many questions in the raw dataset were unsuitable, resulting in a much smaller filtered dataset of 5,516 questions compared to our 31,510. Also, in Figure 11, existing platforms offer limited coverage in earlier years (<300 questions per quarter before Q4 2021), making longitudinal analysis difficult. In contrast, our method supports high scalability and retrospective generation, allowing for uniform coverage across the full time range.
> 2. [Concurrent work](https://arxiv.org/abs/2405.08460) collected 2,532 questions from GoodJudgmentOpen to study temporal generalization. However, the trend is difficult to discern due to the limited number of data points (ranging from 2 to 8) for each model (Table 3). In contrast, while their bi-monthly accuracy results exhibit significant fluctuations, our dataset presents a clearer trend of monthly accuracy degradation, providing deeper insights into how LLM performance evolves over time.
>
> Finally, while we do not claim that LLM-generated questions are of inherently higher quality, we believe our automatic QA generation approach offers several key advantages.  If one sources questions from forecasting markets, the dataset update frequency is dependent on whether there are active users. In contrast, our approach enables daily updates, scalability, and more comprehensive event coverage, making it a valuable complement to human-curated forecasting benchmarks.
> ***
> ### RAG results with one-year retrieval window
> We conduct the suggested experiment with a fixed one-year retrieval window on a subset of 1,500 TF questions (randomly selected 25 questions for each month). As shown in [Fig.S5](https://imgur.com/a/sB7jb1D), our key findings remain consistent: while models can benefit from retrieving more updated articles (RAG cutoff 2024-03), a degradation trend still persists over time. This reinforces the broader conclusion that temporal distance from pretraining continues to impact performance, even with external knowledge augmentation.
> ***
> ### Choice of models in QA generation
> Our initial version of the dataset was generated before GPT-4o became available. We compared question generation across GPT-3.5 & GPT-4 and the newer GPT-4o & GPT-4o-mini models using the same set of articles. Each model produced 48 TF and 48 MC questions, manually evaluated using the same seven QA filtering criteria. Newer models outperformed the older ones with a 54.55% win rate, highlighting their potential to improve question quality in future dataset iterations.
> ***
> ### Others
> 1. Complete RAG results can be found in Appendix B.5.
> 2. Number of articles available for retrieval - [Fig.S6](https://imgur.com/a/qsYFhLT)
> 3. Paleka (2025) introduces a forecasting dataset to test LLMs’ logical coherence. While “generate-then-verify” via LLMs as a common approach, our main contribution lies in capturing and quantifying degradation patterns.

---

> > ### Comment · Reviewer_s4SM · 2025-04-04
> >
> > Thanks for your response. I appreciate the new figures with brier score and fixed retrieval window. The latter was a bit hard to parse because the line with 1 year retrieval window is not labelled. Is it the green one? In this case, any idea why limiting to a 1 year retrieval window actually improves performance for models?
> >
> > I am not particularly satisfied with the quality filtering done in the paper, i.e. LLM based filtering, with some grounding provided by comparing human judgements for filtering. First, It's not clear who these humans are, and why their judgement about forecasting questions (which is quite hard) is reliable. Second, even if the human judgements about filtering were reliable, it still does not provide a good way to measure question quality. Moreover, can the authors compile a comprehensive list of limitations they know about the questions in this dataset, and include it in the paper?
> > Further, the arguments about not using forecasting market data seem a bit hand wavy. In particular, I don't know if raw questions from platforms without filtering are any worse than the LLM generated ones in this dataset. Thus, it's unclear to me why they cannot be used to analyse whether the observed trends are consistent. With LLM generated questions, there could be added unknown confounders, such as question hardness varying with time.
> > I am increasing my score from 2 -> 3, as I think the paper is interesting enough to be worthy of acceptance. I encourage the authors to answer my remaining questions in the original review, as well as some raised here, and if the responses are satisfactory, I am open to increasing the score by another point.
> >
> > **Update based on response to this comment**:
> >
> > The follow up response of the authors clarifies most of my questions and concerns. I will upgrade my score to 4. I think the paper proposes a very useful idea, generating synthetic samples using an LLM grounded in daily updating real-world news. It uses it to show an interesting trend: model performance degrades over time.
> >
> > The only reason I will not go to 5 is that clear quality metrics of the generated samples are not defined, so it's still not clear how to measure progress in this direction. Still, it's definitely a paper with useful insights, worthy of acceptance.

---

> > > ### Author Response · Authors · 2025-04-06
> > >
> > > We thank the reviewer for continued engagement and thoughtful comments.
> > > ### 1. 1-year retrieval window figure
> > > Apologies for the confusion. We provide [Fig. S9](https://imgur.com/a/4xUzOTq), which overlays the original and 1-year window results. The patterns vary by model:
> > > - Qwen-2-7B: no clear preference observed
> > > - LLaMA-3-8B: the 1-year retrieval window often outperforms the full window, as it avoids outdated but semantically similar articles that can mislead the model. This suggests the value of future research on balancing semantic relevance and recency in retrieval.
> > >
> > > We also test the dense retriever all-MiniLM-L6-v2 and find its performance comparable to BM25. While further gains are possible with hybrid retrieval or reranking, we leave it for future work. Our use of BM25 aligns with the choices in prior work (ForecastQA, AutoCast, and TLB-forecast), providing a valid temporal trend.
> > > ### 2. Forecasting market questions
> > > [Prior work](https://arxiv.org/abs/2402.18563) sourced 50,343 [raw questions](https://huggingface.co/datasets/YuehHanChen/forecasting_raw) from 5 forecasting platforms, of which 21,149 are resolved. Among these, 83% are TF, 13% are MC, and others are free-response or numerical. Only 5,516 TF questions remained in their [final dataset](https://huggingface.co/datasets/YuehHanChen/forecasting) after filtering. We find that performance trends on market dataset are noticeably more volatile and harder to interpret compared to ours.
> > >
> > > **a) Lower quality in raw questions**
> > >
> > > Our manual inspection confirms the raw dataset contains much low-quality data, as mentioned in their work. E.g.,
> > > - Will I have a chess.com rating of >1300 ...? (personal)
> > > - Will Jamaica beat Mexico? (no time element)
> > > - Are there more disadvantages in AI than advantages? (ill-defined)
> > >
> > > Of 50 randomly sampled questions, only 28% are well-defined. 26% lack a clear time element, 20% are overly personal, and 26% are ill-defined. Notably, they retain just 5,516 out of 17,477 resolved TF questions—a low acceptance rate (32%) that aligns with our observations.
> > >
> > > **b) Limited earlier year coverage**
> > >
> > > [Fig.S10](https://imgur.com/a/h9QFTJA) shows that the coverage before 2022-10 is sparse, averaging only ~40 raw and ~26 filtered questions per month. This scarcity limits the feasibility of longitudinal trend analysis. In contrast, our method supports high scalability and retrospective generation.
> > >
> > > **c) Harder-to-discern trends using forecasting market questions**
> > >
> > > We run evaluations on TF questions with the forecasting market dataset. The original data is imbalanced, with 61% “No” answers in the raw set and 64% in the filtered set. After balancing, we retain 12,438 questions in raw data and 3,232 in filtered set.
> > >
> > > Fig.S10 shows the model accuracy fluctuates significantly over time. This likely results from several factors:
> > > - **Lower question quality in raw data:** Around 70% raw questions have relatively low quality. Although the dataset size is similar to ours (13,744 in ours vs. 12,438 in raw market data), the quality gap introduces more noise, making trends less stable.
> > > - **Limited early coverage:** Even in the filtered dataset, limited early coverage and inconsistent data volume introduce high variance, reducing the reliability of trend analysis.
> > > - **Confounding factors:** We argue that market questions introduce more confounding factors. [Fig.S11](https://imgur.com/a/jeH44UE) shows the distribution of data sources and question categories varies significantly across time (e.g. more sports-related questions in later periods). Human-written questions also may differ widely in style and difficulty, making them harder to control for consistency. In contrast, our dataset maintains relatively stable distributions over time (see response to Reviewer xifq - 2nd point). While it's theoretically possible to balance the forecasting market dataset, it would reduce the usable size to ~only 300 questions.
> > >
> > > Therefore, our dataset is better suited for revealing performance trends over time due to its scalability, more uniform style and category distribution, and fewer human-introduced confounders.
> > >
> > > ### 3. Question quality
> > > We acknowledge that evaluating valid forecasting questions involves some subjectivity. For human evaluation, 4 graduate students (majoring in finance, accounting, statistics, and data science) from the authors' institution rate the questions using the same detailed instructions given to the model. We see a reasonably consistent inter-human agreement. As LLM-generated data inevitably includes some noise, we provide a [table](https://imgur.com/a/zOvzi0g) summarizing limitations from 100 randomly sampled questions. Most issues fall into categories our filters target, though not perfectly. Still, 83% are valid. Moreover, if the questions were broadly ill-defined, we would not expect to see clear accuracy differences across models or a smooth, consistent degradation over time (e.g. we see a noisy trend in the raw forecast market dataset).

---

### Official Review · Reviewer_xifq · 2025-03-16

**Overall Recommendation:** 2

**Summary:**

This paper proposes a benchmark dataset for assessing a model’s generalization ability in predicting future events and analyzes how model performance evolves over time. Specifically, it compares model performance under three conditions: no access to external information, access to retrieved recent news articles, and access to gold articles. The experimental results indicate that LLMs' prediction accuracy exhibits a significant, gradual decline over time.

**Claims And Evidence:**

1. The paper claims that they conducted a human evaluation to assess the quality of the constructed dataset. However, the evaluation consists of only 60 questions, which may introduce bias into the assessment.
2. In Table 3, the conclusion that model performance degrades over time due to increasing temporal distance from pretraining is not strongly supported. The study does not address whether the distribution of article types, question types, and difficulty levels remains consistent across different years.

**Essential References Not Discussed:**

No

**Experimental Designs Or Analyses:**

1. How should LLMs of different sizes be selected for the experiments in Table 3? The study experiments with models ranging from 2.7B to 7.8B and 56B but seems to lack mid-sized LLMs, such as LLaMA 2-13B or Falcon-40B.

**Methods And Evaluation Criteria:**

1. Human evaluation in section 3.3 is insufficient, making it difficult to ensure dataset quality.

**Other Comments Or Suggestions:**

N/A

**Other Strengths And Weaknesses:**

### Strengths
1. The proposed benchmark can be continuously updated, ensuring long-term usability.

### Weaknesses
1. Constructing a benchmark dataset could be interesting and valuable to the ML community, but it alone is not a sufficient major contribution to ML research, especially given that the quality of the benchmark data is verified with limited human annotation.

**Questions For Authors:**

1. Would incorporating Chain-of-Thought reasoning improve model performance in future event prediction?
2. How does the distribution of question types and difficulty levels vary across different years? Could this affect the observed performance trends?

**Relation To Broader Scientific Literature:**

The paper provides a benchmark that allows continuous updates for evaluating models'
generalization in future event prediction.

**Theoretical Claims:**

N/A

---

> ### Author Rebuttal · Authors · 2025-04-01
>
> We appreciate your thoughtful feedback, and hope to address the concerns below:
> ***
> ### Sample size of human evaluation
> While the evaluation involved 60 questions, we respectfully note that **this sample size aligns with standard practices in similar dataset validation studies**. For example, *TLB-forecast* conducted human evaluations using 24 samples, *SituatedQA* utilized 100 samples for assessing human performance, and *FreshQA* similarly performed human evaluations on 100 samples. To further mitigate bias, we had 4 annotators per question and conducted inter-annotator agreement analyses, as shown in Appendix A.4.
>
> ***
> ### Distribution of question categories and question types
> We provide the distributions of question categories and question types over time in [Fig.S1](https://imgur.com/a/GdqLtRq). Additionally, we conducted further analyses by balancing the categories monthly (selecting TF 5,520 out of 16,783; MC 4,680 out of 14,727) and further **balancing both categories and question types** within MC questions (2,400 out of 5,520). The performance **degradation pattern consistently persists**, showing that degradation primarily arises from increasing temporal distance rather than shifts in category or question distributions. For difficulty-level analysis, while it would be possible to assign difficulty using language models, we believe such automated measures would introduce unnecessary noise. We would rather see the observed model performance itself effectively reflects the inherent difficulty of the questions.
> ***
> ### Evaluation of mid-size LLMs
> We selected models based on recent popular choices, covering both open-source and closed-source options. We appreciate the reviewer’s suggestion regarding mid-sized LLMs. Accordingly, we've included **Llama-2-13B and Qwen-2.5-14B**, and provided an updated plot ([Fig.S2](https://imgur.com/a/BnKiI6t)) and table. As an earlier-generation model, Llama-2-13B underperforms compared to Llama-3-8B, though it still demonstrates higher performance before the knowledge cutoff than after. For Qwen-2.5-14B, we observe relatively strong performance on MC questions but near-random accuracy (~50%) on TF questions. Interestingly, we note that Qwen-2.5-14B exhibits a strong bias towards responding "No," selecting this answer 91.66% of the time on TF questions.
>
> *Table: Yearly Accuracy and YoY Accuracy Change for Llama-2-13B and Qwen-2.5-14B*
> |||K-Cutoff|2020|2021|2022|2023|2024|Pre-Cutoff YoY Change|Post-Cutoff YoY Change |Avg YoY Change|
> |--|-|-|-|-|--|---|--|--|--|--|
> |TF|Llama-2-13B|Sept 2022|56.80|58.59|54.29|51.95|52.65|-0.79%|-8.52%|-1.75%|
> |TF|Qwen-2.5-14B|Unknown|54.02|52.48|52.11|51.74|51.36|-0.99%|-|-0.99%|
> |MC|Llama-2-13B|Sept 2022|42.24|42.31|39.35|37.53|38.74|-1.37%|-12.22%|-1.53%|
> |MC|Qwen-2.5-14B|Unknown|56.54|59.13|56.59|54.60|52.85|-1.38%|-|-1.38%|
>
> ***
> ### Contribution to ML research
> We believe much progress in ML research benefits from open benchmarks. To name a few, ImageNet, GLUE, SQuAD, …. Specifically, our benchmark provides contributions in two important research directions:\
> **(1) LLM forecasting:** Our large-scale, daily-updated dataset reflecting real-world events enables the training and evaluation of models to better support human decision-making.\
> **(2) Continual learning:** Our dataset highlights the challenge of maintaining up-to-date knowledge in LLMs. Our analysis demonstrates that even when provided with gold articles, performance degradation persists, emphasizing the necessity of continuous pre-training to mitigate outdated representations. With Daily Oracle, one could explore how continuous pre-training and efficient adaptation can address the performance degradation challenges presented in our work.
> ***
> ### Would CoT prompting help?
> We randomly sample 25 MC questions per month (1,500 in total) to study how CoT impacts the performance (See [Fig.S3](https://imgur.com/a/0Z9goAA)). For LLaMA-3-70B, we prompt the model to explicitly generate a rationale before providing the final answer. Compared to directly answering, we observe a slight performance improvement initially, though this advantage diminishes after 2023. Additionally, for DeepSeek-R1-Distill-Llama-8B, we utilize the original prompt without modification; however, since this model is fine-tuned to naturally generate reasoning prior to the final answer, we treat its outputs as CoT results. **Its performance similarly matches that of the non-CoT approach.**\
> \
> This observation aligns with recent findings [4], indicating that CoT reasoning primarily benefits tasks involving mathematical, logical, or algorithmic reasoning, with limited gains on other task types. While improved CoT prompt engineering might yield better performance, the results presented here provide baseline insights, leaving further optimization for future research.\
> \
> [4] Sprague, Zayne, et al. "To cot or not to cot? chain-of-thought helps mainly on math and symbolic reasoning." arXiv preprint arXiv:2409.12183 (2024).

---

### Official Review · Reviewer_2v6q · 2025-03-17

**Overall Recommendation:** 4

**Summary:**

The authors propose a method of constructing a continuous temporal knowledge & temporal prediction efficacy benchmark for LLMs. They show results of an implementation of the benchmark, and they describe the release of the benchmark for public use.

**Claims And Evidence:**

Yes, to the best of my knowledge.

**Essential References Not Discussed:**

I did not think of any essential references that were missing.

**Experimental Designs Or Analyses:**

Yes, the authors' submission depends critically on experimental design, specifically in their experiment setup for generating insights from their benchmark. The authors expose interesting aspects of their benchmark by breaking down the task scenario into three settings:

1. Closed-book QA: the LLMs are challenged to answer the temporal QA questions without retrieved documents, i.e. predict the future form their own internal knowledge
2. Constrained open-book QA: the LLMs get access to RAG subject to a retrieval cutoff date
3. Gold open-book QA: the LLMs get access to the exact documents that contain the answer to the question

The authors also carefully track moving averages across the analyzed timespan subject to the various knowledge cutoffs. This allows the authors' study to expose sources of the models' (in)accuracies, e.g. degradation of entity representations over time (parametric knowledge) vs the ability to reason about current information in the context (RAG documents and cutoff).

**Methods And Evaluation Criteria:**

The proposed methods are quite suitable for the problem at hand. The authors describe a convincingly-comprehensive approach to the problem of constructing an automatically-updating continuous benchmark, including:

1. Sourcing from a reputable and established corpus of news on the web (common crawl)
2. Robust filtering procedures
3. Methods to identify truly "current" news rather than opinion pieces discussing past news
4. Methods to extract both multiple-choice and true/false QA questions from current news documents

**Other Comments Or Suggestions:**

No other comments/suggestions

**Other Strengths And Weaknesses:**

I think it's a great benchmark and experimental paper

**Questions For Authors:**

No other questions

**Relation To Broader Scientific Literature:**

The authors' contribution is related to the temporal QA and forecasting areas. Largely, the paper introduces an impactful benchmark to these spaces. It is the first continuously-updating, daily LLM forecasting benchmark, as shown in Table 1.

**Theoretical Claims:**

No theoretical claims were made

---

> ### Author Rebuttal · Authors · 2025-04-01
>
> Thank you for your recognition of our work!

---

### Decision · Program_Chairs · 2025-05-01

**Decision:**

Accept (poster)

**Comment:**

This paper introduces Daily Oracle, a large-scale, daily updating benchmark sourced from real-world news for evaluating LLM temporal knowledge and forecasting capabilities over a long period (2020-2024). The work uses closed-book, RAG, and gold-article settings to dissect LLM performance, revealing clear degradation patterns over time, particularly concerning knowledge cutoffs.

The reviewers overall found the benchmark to be useful, distinguished by its daily generation of forecasting questions directly from news articles over a multi-year period, making it the first continuously updating daily resource of its kind for this task. They also appreciated the insights and analysis of LLM weaknesses. For instance, the benchmark helps differentiate between performance loss due to outdated internal parametric knowledge versus limitations in reasoning ability even when provided with current context (as seen in RAG settings). The persistence of degradation with RAG suggests issues beyond just missing facts, potentially involving outdated representations affecting reasoning.

The most prominent concern was about the quality and reliability of the LLM-generated forecasting questions, supported by reviewer examples of noise and ambiguity. The paper however does validate the approach on a small human evaluation sample (60 questions), which I feel is enough to get a feel for the data. Also, the interpretation of performance degradation, i.e. sharp drops post-knowledge cutoff was contested. The concern was if this degradation reflects a true failure of forecasting/temporal generalization or intended model behavior due to alignment training against speculation. Initial analyses also lacked controls for potential confounders (e.g., impact of varying retrieval index size in RAG, question characteristic shifts over time) and comparison to alternative data sources like forecasting markets. However much of these points were addressed during rebuttal and I hope the authors include some aspects of this discussions to improve the thoroughness of the paper for the next version.